# Inherited basis of visceral, abdominal subcutaneous and gluteofemoral fat depots

Saaket Agrawal [1,2,3,10], Minxian Wang [1,2,4,10], Marcus D. R. Klarqvist [5], Kirk Smith[1,2,3], Joseph Shin[1,2], Hesam Dashti[1], Nathaniel Diamant [5], Seung Hoan Choi [1,2], Sean J. Jurgens [1,2,6], Patrick T. Ellinor[1,2,3], Anthony Philippakis[5,7], Melina Claussnitzer[1,2,3], Kenney Ng[8], Miriam S. Udler [1,2,3], Puneet Batra [5] & Amit V. Khera [1,2,3,9 ✉]

For any given level of overall adiposity, individuals vary considerably in fat distribution. The inherited basis of fat distribution in the general population is not fully understood. Here, we study up to 38,965 UK Biobank participants with MRI-derived visceral (VAT), abdominal subcutaneous (ASAT), and gluteofemoral (GFAT) adipose tissue volumes. Because these fat depot volumes are highly correlated with BMI, we additionally study six local adiposity traits: VAT adjusted for BMI and height (VATadj), ASATadj, GFATadj, VAT/ASAT, VAT/GFAT, and ASAT/GFAT. We identify 250 independent common variants (39 newly-identified) associated with at least one trait, with many associations more pronounced in female participants. Rare variant association studies extend prior evidence for *PDE3B* as an important modulator of fat distribution. Local adiposity traits (1) highlight depot-specific genetic architecture and (2) enable construction of depot-specific polygenic scores that have divergent associations with type 2 diabetes and coronary artery disease. These results – using MRI-derived, BMI-independent measures of local adiposity – confirm fat distribution as a highly heritable trait with important implications for cardiometabolic health outcomes.

[1] Program in Medical and Population Genetics, Broad Institute of MIT and Harvard, Cambridge, MA, USA. [2] Center for Genomic Medicine, Department of Medicine, Massachusetts General Hospital, Boston, MA, USA. [3] Department of Medicine, Harvard Medical School, Boston, MA, USA. [4] CAS Key Laboratory of Genome Sciences and Information, Beijing Institute of Genomics, Chinese Academy of Sciences, Beijing, China. [5] Data Sciences Platform, Broad Institute of MIT and Harvard, Cambridge, MA, USA. [6] Department of Experimental Cardiology, Amsterdam UMC, University of Amsterdam, Amsterdam, The Netherlands. [7] Eric and Wendy Schmidt Center, Broad Institute of MIT and Harvard, Cambridge, MA, USA. [8] Center for Computational Health, IBM Research, Cambridge, MA, USA. [9] Verve Therapeutics, Cambridge, MA, USA. [10] These authors contributed equally: Saaket Agrawal, Minxian Wang. ✉email: avkhera@mgh.harvard.edu

Overall fat mass and fat distribution represent two correlated but distinct axes of variation that determine the health impacts of adipose tissue. Individuals with high body mass index (BMI)—defining obesity—are at elevated risk of type 2 diabetes and cardiovascular events, but increased cardiometabolic risk has also been noted in individuals with the same BMI when fat is disproportionally depleted in more favorable gluteofemoral fat depots and deposited instead in visceral and ectopic fat depots[1–5]. An extreme example of this paradigm occurs in Mendelian lipodystrophies, such as those caused by missense mutations in the *LMNA* and *PPARG* genes[6–10]. By contrast, the genetic architecture of more subtle variation in fat distribution across the general population warrants further attention.

In general, prior studies aiming to elucidate common genetic variation contributing to fat distribution can be categorized into three study types: (1) genome-wide association studies (GWAS) on anthropometric proxies of fat distribution, (2) studies combining GWAS summary statistics of metabolic and anthropometric traits, and (3) GWASs on imaging-based measures of fat distribution. The first type has been spearheaded by the Genetic Investigation of ANthropometric Traits (GIANT) consortium and others, leading to the discovery of over 300 loci associated with waist-to-hip ratio adjusted for BMI (WHRadjBMI) in an analysis of nearly 700,000 individuals[11,12]. Another recent GWAS aimed to examine fat distribution using estimates of body composition based on stepping on a scale equipped with impedance technology, known to be reasonably accurate for total fat volume but less so for fat distribution[13–15]. Despite the considerable value of these studies, a central limitation is an unclear relationship between each anthropometric trait and each fat depot of biological interest—for example, an increase in WHRadjBMI could be capturing increased visceral adipose tissue (VAT; around the abdominal organs), increased abdominal subcutaneous adipose tissue (ASAT; abdominal fat under the skin), decreased gluteofemoral adipose tissue (GFAT; hip and thigh fat), or some combination of these perturbations[16,17]. Variation in WHRadjBMI could also reflect variation in muscle and bone mass, rather than adipose tissue burden.

A second category of studies has aimed to gain further resolution into anthropometric loci by combining summary statistics of metabolic and anthropometric traits, generating clusters of metabolically favorable and unfavorable loci[18–23]. These studies have succeeded in establishing a common variant basis for metabolically distinct fat depots, with seminal work demonstrating that an insulin resistance polygenic score is associated with lower hip circumference in the general population, and that individuals with familial partial lipodystrophy type 1 (FPLD1) have a higher burden of this polygenic score[19]. Along with their reliance on anthropometric proxies of fat distribution, these studies are limited by their inclusion requirement of nominal significance across multiple metabolic traits which is likely leading to only a fraction of the genetic architecture of fat distribution being described.

Finally, the third category of studies performed GWASs on measurements derived from body imaging[24–29]. These include GWASs of CT-quantified VAT and ASAT in nearly 20,000 individuals, GWASs on MRI-quantified VAT and ASAT, and a GWAS of a predicted VAT trait using several anthropometric traits trained on over 4000 DEXA-measured VAT values[26–29]. These studies have been important for translating insights from anthropometric and metabolic trait GWASs to image-derived measurements of the fat depots of interest, but have been limited by (1) the absence of GFAT, which appears to have a metabolically protective role in contrast to VAT and ASAT, and frequently (2) a reliance on raw, unadjusted fat depot metrics which are highly correlated with both each other and BMI.

In this study, we investigate the common and rare variant genetic architecture of three fat depots as quantified by MRI in up to 38,965 UK Biobank participants. Beyond study of raw VAT, ASAT, and GFAT volumes, we analyze six measures that better reflect local adiposity and fat distribution: VAT adjusted for BMI and height (VATadj), ASATadj, GFATadj, VAT/ASAT, VAT/GFAT, and ASAT/GFAT. We show that these local adiposity traits (1) highlight depot-specific genetic architecture, (2) reflect sex-dimorphism previously appreciated with anthropometric traits, and (3) can be used to construct depot-specific polygenic scores that have divergent associations with type 2 diabetes and coronary artery disease. This study is to our knowledge the largest imaging-based study to date to disentangle the genetic architecture of different fat depots—including GFAT, a fat depot that appears to confer protection from adverse cardiometabolic health[5,30].

## Results

VAT, ASAT, and GFAT volumes were quantified in participants of the UK Biobank using a deep learning model trained on body MRI imaging, as previously described (Fig. 1, Supplementary Fig. 1, and Supplementary Table 1)[5]. Among those with MRI-quantified fat depot volumes, 39,076 had genotyping array data available, enabling common variant association studies in up to 38,965 participants after quality control ("Methods"). Mean age in the genotyped cohort was 64.5 years, 51% were female, and 87% were of white British ancestry as previously defined in this study (Supplementary Data 1 and 2). As expected, significant sex differences in fat depot volumes were observed—male participants had higher mean VAT volume (5.0 vs. 2.6 L), while female participants had higher ASAT volume (7.9 vs. 5.9 L) and GFAT volume (11.3 vs. 9.3 L)[31,32].

Six additional adiposity traits—designed to better capture local adiposity—were additionally computed for each individual: VATadj, ASATadj, GFATadj were computed by taking sex-specific residuals against age, age squared, BMI, and height, while VAT/ASAT, VAT/GFAT, and ASAT/GFAT were computed by taking ratios between each pair of fat depots without additional residualization (Supplementary Fig. 5). We tested VATadj, ASATadj, and GFATadj for possible collider bias with BMI or height and found minimal or no evidence of such bias for the majority of genome-wide significant loci (Supplementary Methods, Supplementary Figs. 2–4, and Supplementary Tables 2–5). For example, 87% of VATadj, 86% of ASATadj, and 98% of GFATadj genome-wide significant loci had stronger effect size for the unadjusted fat depot volume compared to BMI, comparable to the 90% of WHRadjBMI loci that met analogous criteria in a recent meta-analysis[12].

In contrast to VAT, ASAT, and GFAT volumes which were highly correlated with BMI (Pearson $r$ ranging from 0.77–0.88), VATadj, ASATadj, GFATadj, and VAT/ASAT were nearly independent of BMI (Pearson $r$ ranging from 0–0.18), while VAT/GFAT (Pearson $r = 0.42$) and ASAT/GFAT (Pearson $r = 0.56$) displayed attenuated correlations with BMI (Fig. 2 and Supplementary Fig. 6A, B). These six derived adiposity traits provided useful, less BMI-dependent metrics for downstream analyses.

**Local adiposity traits are highly heritable and genetically distinct from each other.** To quantify the inherited component to each of these nine adiposity traits, we used the BOLT-REML algorithm to estimate SNP-heritability. Heritability estimates for VAT, ASAT, and GFAT ranged from 0.31–0.36 (standard error (SE) = 0.01), comparable to that observed for BMI in the same individuals ($h_g^2$: 0.31, SE = 0.02)) (Supplementary Table 6). BMI-adjusted fat depots and fat depot ratios tended to have higher heritability compared to unadjusted fat depots and BMI ($h_g^2$ ranging from 0.34–0.41, SE = 0.01–0.02). In contrast,

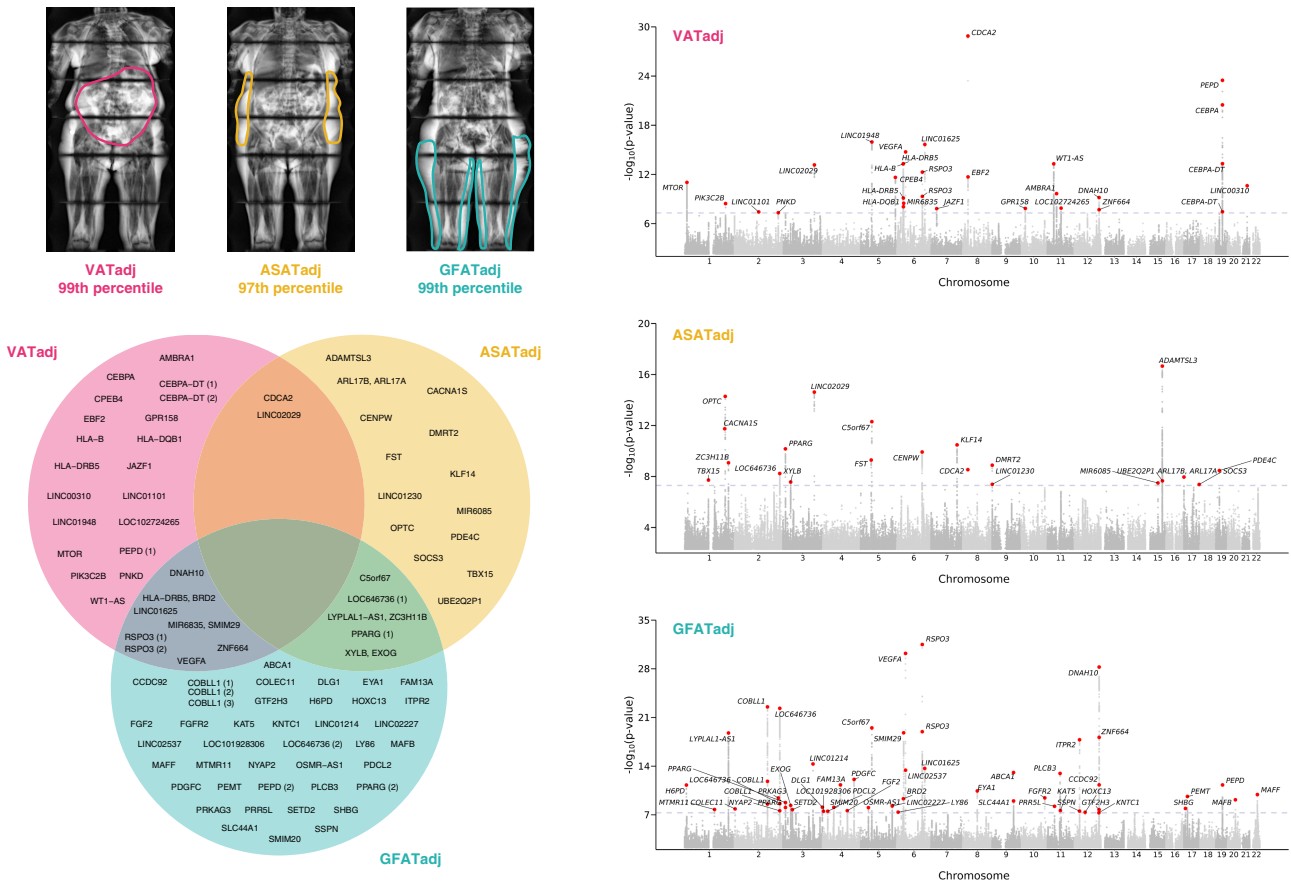

**Fig. 1 Genome-wide association studies of VATadj, ASATadj, and GFATadj.** (Top left) Three female participants from the UK Biobank with similar age (67–70 years) and similar overweight BMI (27.6–28.6 kg/m²) with highly discordant fat distributions (right) Manhattan plots for sex-combined GWASs with VAT adjusted for BMI and height (VATadj), ASATadj, and GFATadj. Lead SNPs are described in Supplementary Data 3. (Bottom left) Overlap between VATadj, ASATadj, and GFATadj loci denoted by the nearest gene; lead SNPs of two traits in high LD ($R^2 \geq 0.1$) were plotted in the intersection. GWAS significance at a commonly used threshold of $p < 5 \times 10^{-8}$ was required for inclusion in the Venn diagram.

WHRadjBMI, an anthropometric proxy for local adiposity, was less heritable than these traits ($h_g^2$: 0.21, SE = 0.01). In sex-stratified analyses, most adiposity traits were more heritable in females as compared to males, with the greatest heritability across all analyses for GFATadj in females ($h_g^2$: 0.52, SE = 0.03).

To study the genetic correlations ($r_g$) between the adiposity and related anthropometric traits, we used LD-score regression[33,34]. Results were generally consistent with observational correlations—raw VAT, ASAT, and GFAT volumes were highly genetically correlated with BMI ($r_g$ ranging from 0.66–0.82), while the three adjusted fat depots, VAT/ASAT, and VAT/GFAT exhibited low genetic correlation with BMI ($r_g$ ranging from −0.16–0.28) (Fig. 2 and Supplementary Fig. 7A, B). In sex-combined analyses, VATadj, ASATadj, and GFATadj were genetically correlated with their unadjusted counterparts ($r_g$ ranging from 0.45–0.59), but nearly independent of the other two fat depots ($r_g$ ranging from −0.24–0.15), suggesting that adjusted-for-BMI traits can enable fat depot-specific genetic analyses. Finally, WHRadjBMI exhibited positive genetic correlations with VATadj ($r_g$: 0.65) and ASATadj ($r_g$: 0.25), and a negative genetic correlation with GFATadj ($r_g$: −0.29), consistent with the perturbations needed in each fat depot to drive a change in WHRadjBMI.

**Common variant architecture of adiposity traits.** We next conducted GWAS for each of the nine adiposity traits—VAT, ASAT, GFAT, VATadj, ASATadj, GFATadj, VAT/ASAT, VAT/GFAT, and ASAT/GFAT—in sex-combined and sex-stratified

groups using BOLT-LMM. After genotyping quality control, we tested associations with 11.5 million imputed SNPs with minor allele frequency (MAF) > 0.005. Across all 27 association studies, 250 loci were associated with at least one adiposity trait at a $p$ value threshold of $5 \times 10^{-8}$ (Supplementary Data 3). If a more stringent genome-wide significance threshold of $5 \times 10^{-9}$ had been used, we would have identified 136 loci, or 85 loci at the most conservative Bonferroni-corrected threshold of $5 \times 10^{-9}/27 = 1.9 \times 10^{-10}$. Of the 250 loci across all adiposity traits, 39 were newly-identified (defined as $R^2 < 0.1$ with all genome-wide significant associations with prior adiposity and relevant anthropometric traits in the GWAS catalog) (Table 1; Methods; and Supplementary Data 4)[35]. Of these 39 loci, 35 have been previously associated with at least one cardiometabolic trait with nominal significance ($p < 0.05$) (Supplementary Table 7). Consistent with heritability estimates, the greatest number of loci were identified in association with GFATadj (54 lead SNPs), while the fewest were identified in association with ASAT (6 lead SNPs). The greatest genomic inflation parameter ($\lambda_{GC}$) was observed with GFATadj ($\lambda_{GC}$: 1.14)—the LD-score regression intercept was 1.05, consistent with polygenicity rather than significant population structure (Supplementary Table 8)[33].

We began by investigating the genetic architecture of VAT, ASAT, and GFAT volumes (Supplementary Fig. 8). All three traits shared a genome-wide significant association with an intronic *FTO* variant (rs56094641) previously associated with childhood and adult obesity[36–38]. ASAT harbored the most significant association with this locus ($p = 1.3 \times 10^{-22}$), followed

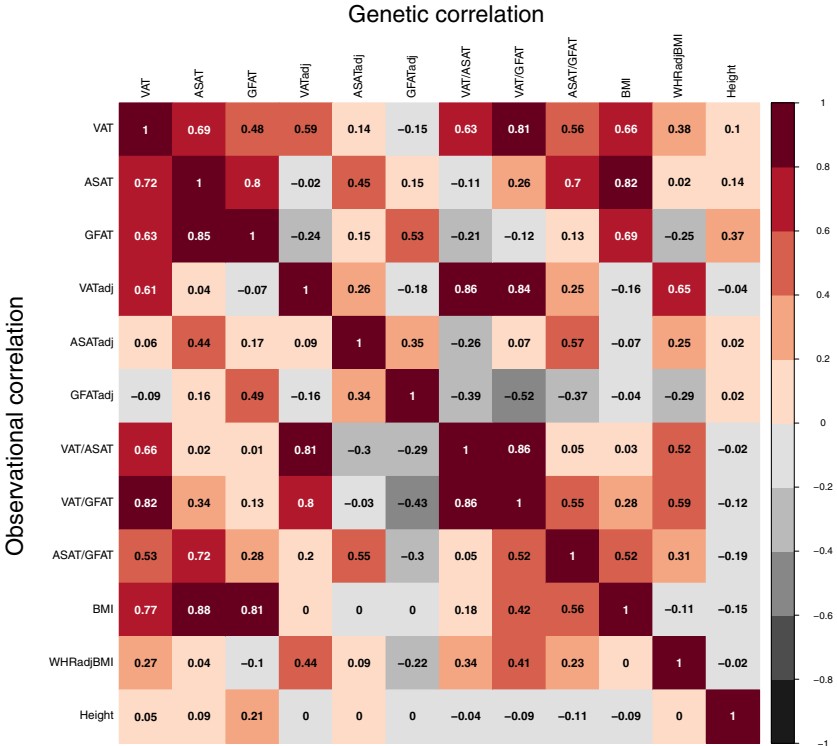

**Fig. 2 Observational and genetic correlations between MRI-derived adiposity traits, BMI, and WHRadjBMI.** Observational correlations displayed are Pearson correlation coefficients. Genetic correlations were obtained from cross-trait LD-score regression using sex-combined summary statistics. Additional correlogram entries, including sex-stratified analyses, are available in Supplementary Figs. 6 and 7.

by GFAT ($p = 1.2 \times 10^{-12}$), and finally VAT ($p = 3.3 \times 10^{-10}$), reflecting the strength of observational and genetic correlation of each fat depot with BMI. Given observational and genetic evidence that a large component of each fat depot volume trait is accounted for by BMI—or "overall adiposity"—we focused further common variant analyses to the three adjusted-for-BMI-and-height measures and three fat depot ratios, aiming to study the genetic architecture of "local adiposity."

For VATadj, 30 genome-wide significant associations were identified ($p < 5 \times 10^{-8}$) (Fig. 1 and Supplementary Fig. 9). The two most significantly associated variants were an intronic *CDCA2* variant (rs11992444; $p = 1.3 \times 10^{-29}$) previously associated with WHRadjBMI and serum triglycerides, and an intronic *PEPD* variant (rs10406327; $p = 3.3 \times 10^{-24}$) previously associated with waist circumference adjusted for BMI (WCadjBMI) and type 2 diabetes[12,39–41]. Newly-identified loci in association with VATadj included an intronic *GPR158* variant (rs1329254; $p = 1.4 \times 10^{-8}$), and an intronic *ARHGEF3* variant exclusively in females (rs1500714; $p = 1.8 \times 10^{-8}$). Prior work has similarly noted female-specific effects of variation in this gene including an association with postmenopausal osteoporosis in humans and *Arhgef3*-KO mice being found to have improved muscle regeneration following injury, with an enhanced rate in females, although the role of this gene on fat distribution is uncertain[42,43].

The most statistically significant association with ASATadj was an intronic *ADAMTSL3* variant (rs768397327; $p = 2.2 \times 10^{-17}$), which was in near-perfect linkage disequilibrium ($R^2 = 0.97$) with another intronic *ADAMTSL3* variant (rs11856122) previously associated with bioelectrical impedance-derived arm fat ratio, leg fat ratio, and trunk fat ratio (Fig. 1 and Supplementary Fig. 10)[13]. Another genome-wide significant signal was observed with an intronic *PPARG* variant (rs527620413). Rare variants in *PPARG* have previously been associated with familial partial lipodystrophy[6,7]. The minor alleles at this locus (MAF = 0.12),

which additionally consisted of rs17036328 and rs71304101 ($R^2 > 0.90$), were associated with increased ASATadj (rs527620413; beta = 0.071; $p = 6.8 \times 10^{-11}$), increased GFATadj (rs71304101; beta = 0.062; $p = 1.7 \times 10^{-9}$), decreased VAT/ASAT ratio (rs17036328; beta = −0.080; $p = 5.8 \times 10^{-15}$), and decreased VAT/GFAT ratio (rs17036328; beta = −0.058; $p = 2.4 \times 10^{-8}$). These three SNPs are also in high LD ($R^2 \geq 0.94$) with rs1801282, a missense variant in *PPARG* previously associated with reduced risk of type 2 diabetes[44–46]. These data suggest that common variation at *PPARG* can lead to adiposity variation along the lipodystrophy axis—for this locus, the minor alleles associated with a pattern of favorable adiposity. *FST* is another gene that promotes adipogenesis and may have a causal role in insulin resistance—an intronic variant in *FST* (rs55744247) associated with ASATadj ($p = 5.1 \times 10^{-10}$), but not VATadj ($p = 0.80$) or GFATadj ($p = 0.25$)[47]. Finally, a newly-identified intronic *DMRT2* variant (rs6474550; $p = 1.3 \times 10^{-9}$) associated with ASATadj. In a study investigating fat depot-specific transcriptome signatures before and after exercise, *DMRT2* was one of three genes with higher expression in ASAT vs. GFAT both before and after exercise[48].

The top GFATadj signal was an intronic *RSPO3* variant (rs72959041; $p = 3.2 \times 10^{-32}$) that has previously been shown to be a top signal for WHRadjBMI (Fig. 1 and Supplementary Fig. 11)[12]. Recent work clarified this SNP as the causal variant at the locus and suggested that the minor allele concurrently reduces leg fat mass and increases android fat mass[49]. Our results confirm and further clarify these findings—the minor allele (MAF = 0.05) associated with marked reduction of GFATadj (beta = −0.195; $p = 3.2 \times 10^{-32}$) and increased of VATadj (beta = 0.118; $p = 7.8 \times 10^{-13}$), but a nonsignificant effect on ASATadj (beta = −0.029; $p = 0.09$). Three independent intronic *COBLL1* variants ($R^2 < 0.1$) were associated with GFATadj (rs13389219; $p = 3.0 \times 10^{-23}$, rs3820981; $p = 1.5 \times 10^{-12}$, rs34224594;

**Table 1 Forty-two newly-identified locus-trait associations in this study.**

| Trait | CHR | BP | SNP | Effect allele | Other allele | EAF | BETA | SE | p value | Nearest gene |
|---|---|---|---|---|---|---|---|---|---|---|
| GFAT | 11 | 95840436 | rs1074742 | A | G | 0.401 | 0.041 | 0.007 | 1.40E−08 | MAML2 |
| GFAT | 12 | 124344710 | rs138756410 | T | C | 0.986 | −0.172 | 0.031 | 3.00E−08 | DNAH10 |
| GFAT | 12 | 125092343 | rs4765159 | A | G | 0.018 | 0.146 | 0.027 | 3.50E−08 | NCOR2 |
| VATadj | 2 | 121310704 | rs35932591 | C | T | 0.879 | 0.061 | 0.011 | 3.80E−08 | LINC01101 |
| VATadj | 10 | 25767521 | rs1329254 | C | T | 0.37 | 0.042 | 0.007 | 1.40E−08 | GPR158 |
| VATadj | 11 | 69195097 | rs7933253 | T | C | 0.048 | 0.098 | 0.017 | 1.30E−08 | LOC102724265 |
| VATadj | 2 | 121310704 | rs35932591 | C | T | 0.88 | 0.086 | 0.016 | 3.90E−08 | LINC01101 |
| VATadj (Male) | 3 | 56901687 | rs1500714 | C | G | 0.854 | 0.081 | 0.015 | 1.80E−08 | ARHGEF3 |
| ASATadj | 1 | 201016296 | rs3850625 | G | A | 0.882 | −0.079 | 0.011 | 1.80E−12 | CACNA1S |
| ASATadj | 9 | 1044400 | rs2048235 | C | T | 0.384 | 0.041 | 0.007 | 4.10E−08 | LINC01230 |
| ASATadj | 9 | 1052722 | rs6474550 | G | T | 0.66 | 0.045 | 0.008 | 1.30E−09 | DMRT2 |
| ASATadj | 15 | 62757857 | rs17205757 | A | G | 0.674 | −0.042 | 0.008 | 3.20E−08 | MIR6085 |
| ASATadj | 17 | 76324751 | rs4444401 | A | G | 0.473 | −0.04 | 0.007 | 4.20E−08 | SOCS3 |
| ASATadj (Female) | 1 | 116916645 | rs749166380 | CT | C | 0.102 | 0.102 | 0.018 | 2.20E−08 | ATP1A1 |
| ASATadj (Female) | 8 | 58352327 | rs776481989 | ATAAT | A | 0.998 | 0.795 | 0.134 | 8.60E−09 | LOC101929488 |
| GFATadj | 2 | 3648186 | rs7588285 | C | G | 0.188 | 0.053 | 0.009 | 1.40E−08 | COLEC11 |
| GFATadj | 2 | 226768344 | 2:226768344_CA_C | CA | C | 0.193 | −0.051 | 0.009 | 2.60E−08 | NYAP2 |
| GFATadj | 3 | 196818853 | rs13099700 | A | G | 0.722 | 0.047 | 0.008 | 7.90E−09 | DLG1 |
| GFATadj | 5 | 38810354 | rs142369482 | G | GT | 0.656 | −0.044 | 0.008 | 9.10E−09 | OSMR-AS1 |
| GFATadj | 10 | 122970216 | rs1907218 | T | C | 0.314 | −0.049 | 0.008 | 3.60E−10 | FGFR2 |
| GFATadj (Male) | 4 | 104780790 | rs528845403 | A | AATGTGT | 0.991 | −0.325 | 0.061 | 2.40E−08 | TACR3 |
| GFATadj (Female) | 1 | 181161153 | rs7550430 | A | G | 0.998 | 0.892 | 0.144 | 1.80E−09 | LINC01732 |
| GFATadj (Female) | 2 | 165533198 | rs386652275 | T | TC | 0.974 | −0.19 | 0.034 | 3.20E−08 | COBLL1 |
| VAT/ASAT | 2 | 178121005 | rs13028464 | C | T | 0.631 | −0.039 | 0.007 | 4.80E−08 | NFE2L2 |
| VAT/ASAT | 6 | 19947871 | rs70987287 | T | TTTTTA | 0.728 | 0.064 | 0.008 | 1.70E−17 | ID4 |
| VAT/ASAT | 8 | 25459001 | rs3890765 | C | A | 0.941 | −0.084 | 0.015 | 6.80E−09 | CDCA2 |
| VAT/ASAT | 9 | 1054362 | rs6474552 | G | C | 0.432 | −0.04 | 0.007 | 1.20E−08 | DMRT2 |
| VAT/ASAT | 10 | 63702572 | rs55767272 | A | C | 0.937 | 0.085 | 0.014 | 6.80E−09 | ARID5B |
| VAT/ASAT | 10 | 122992475 | rs11199845 | C | T | 0.46 | 0.055 | 0.007 | 1.50E−14 | FGFR2 |
| VAT/ASAT | 2 | 61760756 | rs13390751 | A | C | 0.838 | 0.076 | 0.013 | 1.30E−08 | XPO1 |
| VAT/ASAT (Male) | 6 | 19949170 | 6:19949170_GT_G | GT | G | 0.746 | 0.068 | 0.012 | 3.70E−09 | ID4 |
| VAT/ASAT (Male) | 10 | 122992442 | rs11199844 | C | T | 0.463 | 0.059 | 0.01 | 5.90E−09 | FGFR2 |
| VAT/ASAT (Female) | 6 | 19947871 | rs70987287 | T | TTTTTA | 0.729 | 0.064 | 0.011 | 8.50E−10 | ID4 |
| VAT/ASAT (Female) | 12 | 121319417 | rs59757908 | T | C | 0.995 | −0.425 | 0.076 | 4.20E−08 | SPPL3 |
| VAT/GFAT | 14 | 94844947 | rs28929474 | C | T | 0.982 | 0.16 | 0.026 | 4.80E−10 | SERPINA1 |

**Table 1 (continued)**

| Trait | CHR | BP | SNP | Effect allele | Other allele | EAF | BETA | SE | p value | Nearest gene |
|---|---|---|---|---|---|---|---|---|---|---|
| VAT/GFAT (Female) | 1 | 162430821 | rs960318 | G | C | 0.203 | 0.068 | 0.012 | 1.80E-08 | UHMK1 |
| VAT/GFAT (Female) | 2 | 116072770 | rs1399916 | T | TA | 0.256 | 0.06 | 0.011 | 3.70E-08 | DPP10 |
| VAT/GFAT (Female) | 6 | 32975699 | rs9276981 | G | C | 0.809 | −0.064 | 0.012 | 4.60E-08 | HLA-DOA |
| ASAT/GFAT | 5 | 55830865 | rs39837 | C | T | 0.667 | 0.043 | 0.007 | 2.60E-08 | LINC01948 |
| ASAT/GFAT | 14 | 95219657 | rs8006225 | G | T | 0.817 | 0.055 | 0.009 | 2.60E-09 | GSC |
| ASAT/GFAT | 16 | 86424697 | rs1552657 | G | A | 0.549 | −0.037 | 0.007 | 4.90E-08 | LINC00917 |
| ASAT/GFAT (Female) | 5 | 55830865 | rs39837 | C | T | 0.666 | 0.061 | 0.01 | 9.10E-09 | LINC01948 |

Newly-identified loci were defined as loci that associated with an adiposity trait with $p < 5 \times 10^{-8}$ and that were not in LD ($R^2 < 0.10$) with any of the loci in the GWAS catalog for adiposity or related anthropometric traits (see "Methods"). "adj" traits are adjusted for BMI and height (see "Methods"). Note that rs3593591 (VATadj and VATadj (Male)), rs7098287 (VAT/ASAT and VAT/ASAT (Female)), and rs39837 (ASAT/GFAT and ASAT/GFAT (Female)) are duplicated, so 39 unique lead SNPs are presented in this table. Loci were additionally cross-referenced with prior studies using the Type 2 Diabetes Knowledge Portal (Supplementary Table 7).
BP GRCh37 position, EAF effect allele frequency, BETA effect size per effect allele, p value BOLT-LMM association p value.

$p = 2.8 \times 10^{-9}$), but not VATadj ($p_{min} = 0.009$) or ASATadj ($p_{min} = 2.7 \times 10^{-3}$). One of these variants (rs13389219) is in LD with another intronic *COBLL1* variant (rs6738627) which has previously been implicated in a metabolically healthy obesity phenotype characterized by increased HDL cholesterol and reduced triglycerides despite increased body fat percentage[50]. In this study, aligning rs13389219 to the BMI-increasing direction (beta = 0.011, $p = 7.3 \times 10^{-3}$) revealed a concurrent increase in GFATadj (beta = 0.073), consistent with a metabolically healthy fat depot shift. Finally, a GFATadj association was observed at an intronic *PDGFC* variant (rs6822892; $p = 8.0 \times 10^{-13}$)—*PDGFC* was recently prioritized as a candidate causal gene for insulin resistance in human preadipocytes and adipocytes[47].

Several associations were exclusive to GWASs of fat depot ratios (Supplementary Figs. 12–14). A missense variant in *ACVR1C* significantly reduced VAT/GFAT ratio (rs55920843; MAF = 0.01; beta = −0.18; $p = 1.9 \times 10^{-8}$). Prior work demonstrated that sequence variation in *ACVR1C*—including this variant—reduces WHRadjBMI and risk of type 2 diabetes[51]. Another missense variant in *ACVR1C* was nominally associated with reduced VAT/GFAT ratio, strengthening the importance of this gene (rs56188432 (p.Ile195Thr); beta = −0.21, $p = 0.006$) (Supplementary Data 5). Finally, a newly-identified association was present between VAT/GFAT ratio and a missense variant in *SERPINA1* (rs28929474; MAF = 0.02; beta = −0.16; $p = 4.8 \times 10^{-10}$). Homozygous carriers of this variant are known to harbor alpha-1-antitrypsin deficiency, and heterozygous carriers have higher serum ALT and increased risk of cirrhosis[51,52]. Interestingly, this missense variant has also been associated with reduced risk of type 2 diabetes (odds ratio: 0.90, $p = 5.9 \times 10^{-6}$) and coronary artery disease (odds ratio: 0.88, $p = 9.4 \times 10^{-9}$)[41,53]. The present association with reduced VAT/GFAT ratio suggests that a shift toward a metabolically healthy fat distribution could partially explain a reduced risk of cardiometabolic disease. In a large meta-analysis, this *SERPINA1* variant had only a nominally significant association with waist-to-hip ratio (beta = −0.03, $p = 3.4 \times 10^{-4}$)—the closest anthropometric correlate of VAT/GFAT ratio—highlighting the utility of image-derived phenotypes for this discovery[12].

**Gluteofemoral adiposity signal classification**. We aimed to categorize genetic loci associated with gluteofemoral adiposity—postulated to be metabolically protective—into distinct clusters. Starting with the 250 lead SNPs that were associated ($p < 5 \times 10^{-8}$) with any of the nine adiposity traits in this study, we selected 101 LD-pruned ($r^2 = 0.1$) SNPs that were nominally associated ($p < 0.05$) with GFATadj. Each SNP was aligned to the GFATadj increasing direction. We used Bayesian non-negative matrix factorization (bNMF)—a soft clustering approach—with 32 cardiometabolic traits including anthropometric traits (e.g., BMI, body fat percentage), lipid traits (e.g., triglycerides, HDL-cholesterol, and total cholesterol), and diabetes-related traits (e.g., glucose, hemoglobin A1C) to identify clusters (Supplementary Data 6).

In all 100 iterations, the data converged to three clusters (Supplementary Data 7). The most strongly weighted traits for the first cluster included increased HDL-cholesterol, decreased serum triglycerides, decreased hemoglobin A1C, and decreased alanine aminotransferase, consistent with a metabolically healthier fat distribution. Top loci in this cluster included several well-known associations with WHRadjBMI and insulin resistance including *COBLL1*, *RSPO3*, *PPARG*, and *DNAH10*[12,47,54,55]. A second cluster appeared to be related to inflammatory pathways, with top loci including *HLA-DRB5*, *HLA-B*, and *MAFB*—*MAFB* has previously been implicated as a regulator of adipose tissue inflammation[56]. Strongly weighted traits in this cluster included

decreased aspartate aminotransferase, decreased total cholesterol, and decreased C-reactive protein. The third and final cluster appeared to reflect the interplay between hepatocyte biology and fat distribution with top loci including a missense variant in *SERPINA1* and *SHBG*—the former is known to cause alpha-1-antitrypsin deficiency and has been previously associated with increased ALT and cirrhosis, and sex-hormone binding globulin is synthesized by hepatocytes and is reduced in patients with non-alcoholic fatty liver disease[57,58]. Strongly weighted traits in this cluster included increased albumin, increased sex-hormone binding globulin, and increased total protein.

To test the robustness of these results, we performed two sensitivity analyses. First, we performed clustering using 85 LD-pruned SNPs nominally associated ($p < 0.05$) with unadjusted GFAT. The three aforementioned clusters were reproduced along with a fourth cluster representing overall adiposity—the top locus in this cluster was *FTO* and the most strongly weighted trait was increased BMI (Supplementary Data 8). Finally, we performed one additional clustering analysis of the same 101 LD-pruned SNPs for GFATadj, this time including VATadj and ASATadj as clustering traits alongside the 32 previously used cardiometabolic traits, resulting in a nearly identical set of three clusters (Supplementary Data 9).

**Sex heterogeneity in genetic associations with local adiposity traits.** Given prior work has noted significant sex heterogeneity in the genetic basis of anthropometric traits, we next tested for such heterogeneity for each of the six local adiposity traits (VATadj, ASATadj, GFATadj, VAT/ASAT, VAT/GFAT, and ASAT/GFAT)[11,12,55,59]. Genetic correlations between sex-stratified summary statistics indicated overall high correlation between traits, with $r_g$ somewhat higher for VATadj ($r_g = 0.87$) as compared to ASATadj or GFATadj ($r_g = 0.80$ and 0.79 respectively) (Supplementary Table 9). We next tested for sex-dimorphism across loci that were genome-wide significant for either sex-combined or sex-stratified analyses for each local adiposity trait (Fig. 3, Supplementary Fig. 15, and Supplementary Data 10). Three of 34 VATadj loci (9%), six of 27 ASATadj loci (22%), and six of 65 GFATadj (9%) showed significant sex dimorphism ($p_{diff}$ <0.05/220 independent loci-trait pairs tested = $2.3 \times 10^{-4}$). The majority of these signals were driven by a greater magnitude of effect in female participants, which is consistent with prior investigations of WHRadjBMI[12,55]. Across all six local adiposity traits, 26 trait-loci associations were only genome-wide significant in females, while 9 loci were only genome-wide significant in males.

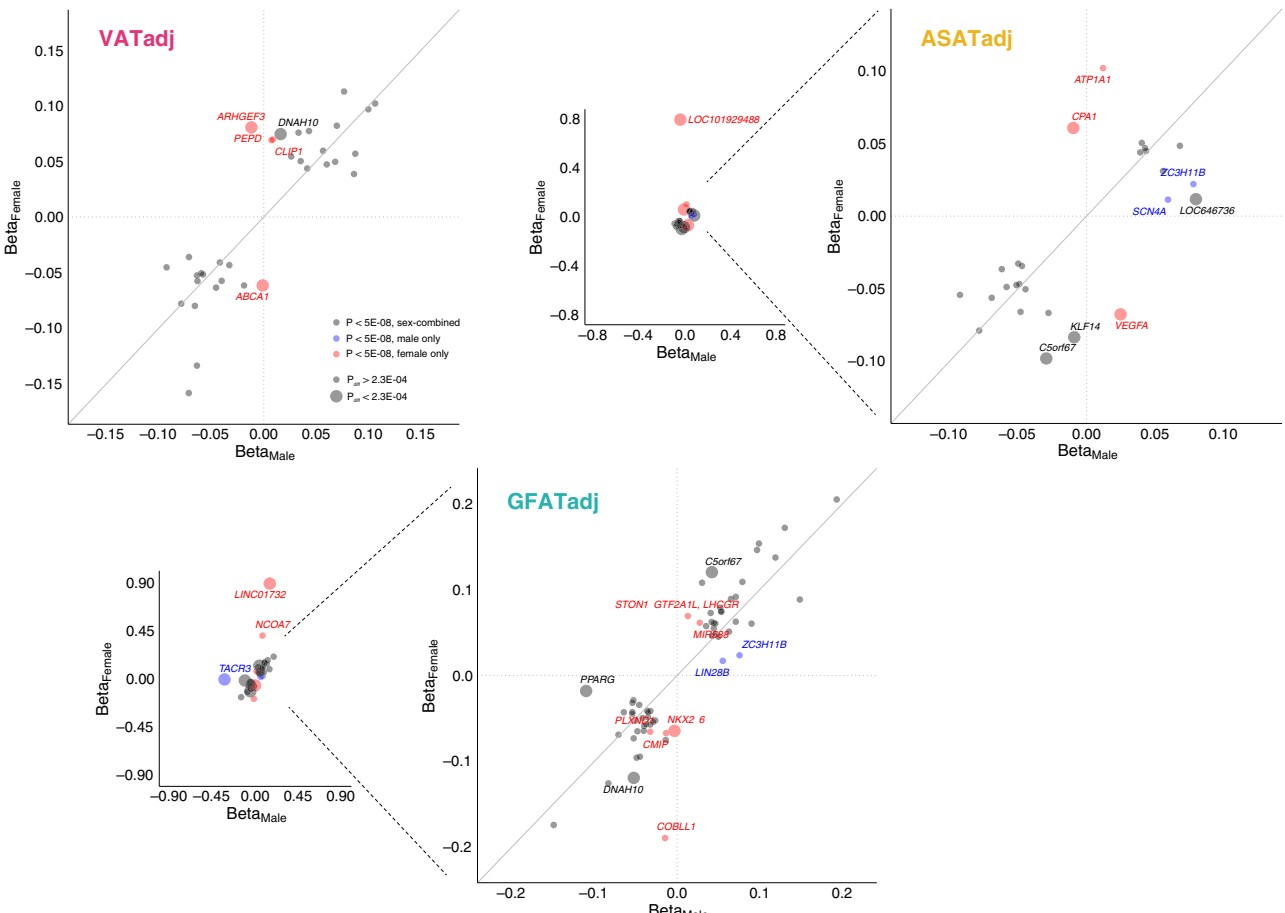

**Fig. 3 Common variant sex heterogeneity for VATadj, ASATadj, and GFATadj local adiposity traits.** For each adiposity trait, independent loci that were associated with the trait in either sex-combined or sex-stratified analyses are plotted (Supplementary Data 10). Thirty-four such loci are plotted for VATadj, 27 for ASATadj, and 65 for GFATadj. Loci colored black were genome-wide significant ($p < 5 \times 10^{-8}$) in sex-combined analysis, blue loci were significant for males, but neither females nor sex-combined, and red loci were significant for females, but neither males nor sex-combined. $p_{diff}$ corresponds to the "calcpdiff" function in EasyStrata comparing SNP effects in males and females (Methods). Across six adiposity traits (VATadj, ASATadj, GFATadj, VAT/ASAT, VAT/GFAT, and ASAT/GFAT), 220 unique loci-trait pairs were tested for sex heterogeneity (Supplementary Fig. 15), so a Bonferroni-corrected significance threshold of $p_{diff} < 0.05/220 = 2.3 \times 10^{-4}$ was set.

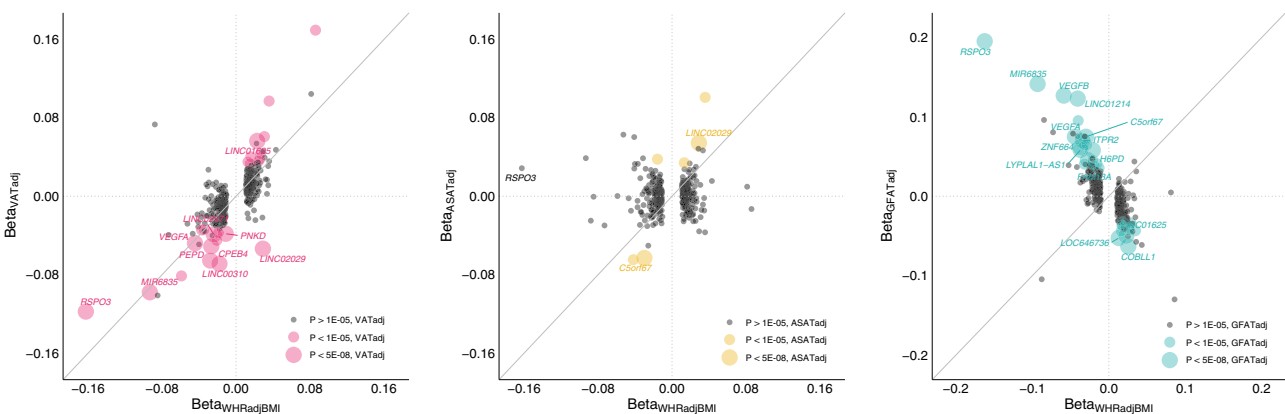

**Fig. 4 Effects of previously identified WHRadjBMI loci on local adiposity traits.** In total, 345 of the 346 index SNPs associated with WHRadjBMI in a recent meta-analysis from the GIANT consortium were available in the studied cohort[12]. Effect sizes of VATadj, ASATadj, and GFATadj are plotted against the effect size for WHRadjBMI as reported in the cited study (Supplementary Data 11). Betas and p values for VATadj, ASATadj, and GFATadj correspond to the BOLT-LMM association p values computed in this study for the 345 index SNPs.

**Overlap of local adiposity traits with WHRadjBMI findings**. To investigate the added value of precisely quantifying fat depots with MRI in a smaller number of individuals as compared to WHRadjBMI in a larger cohort, we studied the effects of 345 loci identified in the most recent WHRadjBMI meta-analysis of up to 694,649 individuals on VATadj, ASATadj, and GFATadj (Fig. 4 and Supplementary Data 11)[12]. Of the 345 loci, 10 (3%) achieved genome-wide significance in association with VATadj ($p < 5 \times 10^{-8}$), 2 with ASATadj (0.6%), and 14 (4%) with GFATadj. A unit increase in WHRadjBMI might be expected to be reflecting a unit increase in VATadj or ASATadj, or a unit decrease in GFATadj. We quantified how often a locus was discordant from this pattern (e.g., a unit increase in WHRadjBMI corresponding to a unit decrease in VATadj), excluding loci where the fat depot effect size was smaller in magnitude than the SE. Fifteen of 242 loci (6%) were VATadj-discordant, 71 of 166 loci (43%) were ASATadj-discordant, and 22 of 231 loci (10%) were GFATadj-discordant (Supplementary Data 11).

Two illustrative examples indicate how follow-up of WHRadjBMI associations from a very large study in a smaller study with specific fat depots quantified may prove useful. The top WHRadjBMI signal is located at an intronic *RSPO3* locus (rs72959041; beta = −0.162; $p = 2.1 \times 10^{-293}$)—our work further clarifies that this signal is driven by an effect on VATadj (beta = −0.118; $p = 7.8 \times 10^{-13}$) and GFATadj (beta = 0.195; $p = 3.2 \times 10^{-32}$), but not ASATadj (beta = 0.029; $p = 0.09$). In contrast, a WHRadjBMI signal near *LINC02029* (rs10049088; beta = 0.029; $p = 1.5 \times 10^{-59}$) is driven by ASATadj (beta = 0.054; $p = 7.3 \times 10^{-14}$) and GFATadj (beta = −0.034, $p = 6.0 \times 10^{-6}$), but has a VATadj-discordant signal (beta = −0.053, $p = 8.7 \times 10^{-13}$).

**External validation**. We pursued replication of our genome-wide significant loci with a prior meta-analysis of CT and MRI-derived VAT, ASAT, VAT adjusted for BMI (VATadjBMI), and VAT/ASAT ratio in up to 18,332 individuals[27]. Of the 76 SNP-trait associations across the traits of VAT, ASAT, VATadj, and VAT/ASAT ratio in this study, association results for 17 were available for comparison in published summary statistics[27]. Of these, 16 (94%) had directionally consistent effects (binomial test $p = 2.7 \times 10^{-4}$, Supplementary Data 12).

**Transcriptome-wide association study**. To prioritize genes, we conducted a transcriptome-wide association study (TWAS) using gene expression data from visceral and subcutaneous adipose tissue from GTEx v7[60]. Across all traits, the most significant association was observed between GFATadj and *CCDC92* (TWAS

Z-score = 12.0; TWAS $p = 2.7 \times 10^{-33}$) in subcutaneous adipose tissue (Supplementary Data 13). The most significant eQTL for this association was shared with *DNAH10OS* (TWAS Z-score = 10.5; $p = 8.2 \times 10^{-26}$) and *DNAH10* (TWAS Z-score = 7.9; $p = 3.5 \times 10^{-15}$). Prior work demonstrated that knockdown of *CCDC92* or *DNAH10* led to significant reduction of lipid accumulation in an adipocyte model[19]. Of note, predicted VATadj associations with *CCDC92* and *DNAH10* in visceral adipose tissue samples demonstrated the opposite direction of effect (*CCDC92* Z-score = −6.7; $p = 2.7 \times 10^{-11}$; *DNAH10* Z-score = −5.3; $p = 1.1 \times 10^{-7}$), suggesting fat depot discordant effects.

Another top TWAS signal was observed with GFATadj and *IRS1* (Z-score = 9.1; $p = 6.2 \times 10^{-20}$) with the corresponding association with ASATadj having the same direction of effect (Z-score = 5.5; $p = 4.6 \times 10^{-8}$). Prior work has demonstrated that decreased *IRS1* expression, the gene encoding the insulin receptor substrate, causes insulin resistance—our work further suggests that impaired expansion of the gluteofemoral and abdominal subcutaneous fat depots may be involved in this physiological insult[47,61]. Finally, a significant association was observed between *VEGFB* and GFATadj (Z-score = 7.0; $p = 2.0 \times 10^{-12}$), but not ASATadj (Z-score = 0.44; $p = 0.66$). Endothelial *VEGFB* is known to facilitate endothelial targeting of fatty acids to peripheral tissues and induce adipocyte thermogenesis, and transduction of *VEGFB* into mice improved metabolic health without changes in body weight[62,63]. These results suggest that maintenance of the gluteofemoral fat depot may partially explain the metabolic effects of *VEGFB*.

**Tissue-specific enrichment analyses**. We used stratified LD-score regression to probe for tissue-specific enrichment for each adiposity trait (Supplementary Data 14)[64]. A marked dichotomy was observed between the three raw fat depot volumes (VAT, ASAT, GFAT)—each highly genetically correlated with BMI—and the six derived local adiposity traits (VATadj, ASATadj, GFATadj, VAT/ASAT, VAT/GFAT, ASAT/GFAT). While VAT, ASAT, and GFAT showed a pattern of central nervous system (CNS) tissue enrichment—consistent with the enrichment pattern for BMI—local adiposity traits were characterized by adipose tissue signals with reduced CNS signals (Supplementary Figs. 16 and 17). These results further emphasize that the genetic basis of overall adiposity is driven largely by CNS processes—such as those governing appetite and satiety—whereas fat distribution is regulated at the level of the adipocyte and other peripheral tissues.

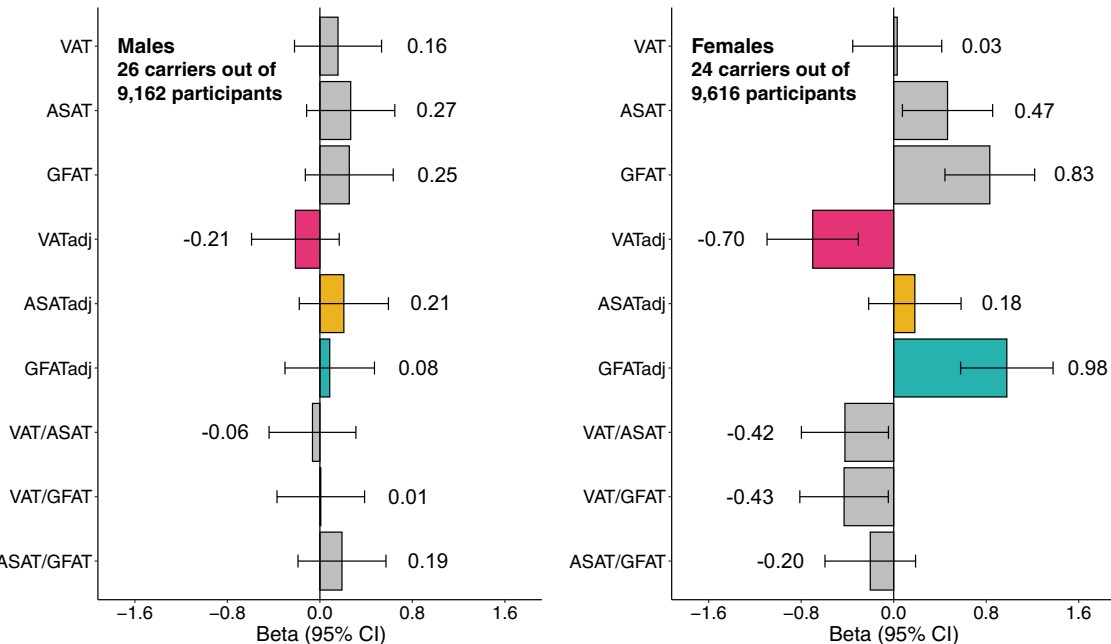

**Fig. 5 Rare variants in *PDE3B* selectively associate with fat distribution in female participants.** A mask combining predicted loss-of-function variants and missense variants predicted to be deleterious by 5 out of 5 in silico prediction algorithms in *PDE3B* associated with GFATadj in females with exome-wide significance (Supplementary Data 15). Effect sizes with 95% confidence intervals are plotted for carrier status. Linear regressions were adjusted for age, age squared, imaging center, genotyping array, and the first ten principal components of genetic ancestry (Supplementary Data 16). Note that the carrier counts are with respect to individuals who had "adj" traits available. For the other six traits, the carrier counts are 26 carriers/9616 participants for males and 25 carriers/9879 participants for females.

**Rare variant association study**. Up to 19,255 individuals with fat depots quantified and exome sequencing data available were included in rare variant association studies. We utilized two masks: one containing only predicted loss-of-function variants (pLoF) and a second combining pLoF with missense variants predicted to be deleterious by 5 out of 5 in silico prediction algorithms (pLoF + missense). We tested the association between the aggregated rare variant score with each mask and each inverse normal transformed phenotype using multivariable regression. Analyses were restricted to genes with at least ten variant carriers in the analyzed cohort, yielding up to 12,020 tested genes. Exome-wide significance was considered to be $p < 0.05/12,020 = 4.2 \times 10^{-6}$, while a Bonferroni-corrected study-wide significance threshold was set to $p < 4.2 \times 10^{-6}/27 = 1.5 \times 10^{-7}$. One exome-wide significant association was identified: pLoF + missense variants in *PDE3B* associated with increased GFATadj in females (24 carriers; beta = 0.98; $p = 1.7 \times 10^{-6}$) (Supplementary Data 15). Individuals who carry loss-of-function variants in *PDE3B* have previously been demonstrated to have reduced WHRadjBMI[65]. This study confirms and extends this result by demonstrating that females who carry pLoF + missense variants in *PDE3B* harbor increased GFATadj and reduced VATadj (beta = −0.70; $p = 5.1 \times 10^{-4}$)—consistent with a metabolically favorable fat distribution—and that these effects are attenuated in males (GFATadj beta = 0.08; $p = 0.67$; VATadj beta = −0.21; $p = 0.27$) (Fig. 5 and Supplementary Data 16).

Rare variant signals in two additional genes, while they did not reach our threshold for exome-wide significance, warrant discussion. pLoF + missense variants in *PCSK1* associated with GFAT in sex-combined analysis (101 carriers; beta = 1.11; $p = 7.5 \times 10^{-6}$) and pLoF + missense variants in *ACAT1* associated with VAT in females (23 carriers; beta = 2.66; $p = 6.4 \times 10^{-6}$). Both of these genes have previously been implicated in altering adiposity. Rare mutations in *PCSK1* are known to cause monogenic obesity—here, a relatively symmetric

pattern of increased GFAT, VAT (beta = 0.87; $p = 4.1 \times 10^{-4}$), and ASAT (beta = 1.04; $p = 3.1 \times 10^{-5}$) were observed in sex-combined analyses (Supplementary Data 16)[66,67]. In a study comparing obese women with or without type 2 diabetes, gene expression of *ACAT1* was downregulated in the VAT and ASAT of obese women with type 2 diabetes and expression was restored after bariatric surgery and weight loss, suggesting a role in obesity-associated insulin resistance[68].

Finally, we investigated if rare variants in known familial partial lipodystrophy genes *PPARG* and *LMNA* were associated with the adiposity traits defined in this study (Supplementary Data 17)[8,10,69]. The 17 carriers of a pLoF + missense variant in *PPARG* tended to have reduced GFATadj in sex-combined analysis (beta = −0.99, $p = 0.05$), consistent with a lipodystrophic-pattern of reduced peripheral adipose tissue deposition. We were unable to detect a significant association among the 51 carriers of rare *LMNA* variants, potentially related to inadequate statistical power or variant annotation.

**Polygenic contribution to extremes of VATadj, ASATadj, and GFATadj**. Because many individuals with lipodystrophy-like phenotypes—especially in its more subtle forms—do not harbor a known pathogenic rare variant, prior studies have begun to explore a potential "polygenic lipodystrophy," in which an inherited component is instead driven by the cumulative impact of many common DNA variants[10,19,20,70]. In the context of the traits defined in this study, a lipodystrophy-like phenotype might be characterized by increased VATadj, decreased ASATadj, and/or decreased GFATadj. We set out to quantify the potential for genetic prediction of these traits by generating polygenic scores consisting of up to 1,125,301 variants for VATadj, ASATadj, and GFATadj traits using the LDpred2 algorithm[71]. To ensure no overlap between summary statistics and tested individuals, GWAS was conducted using a randomly selected 70% of

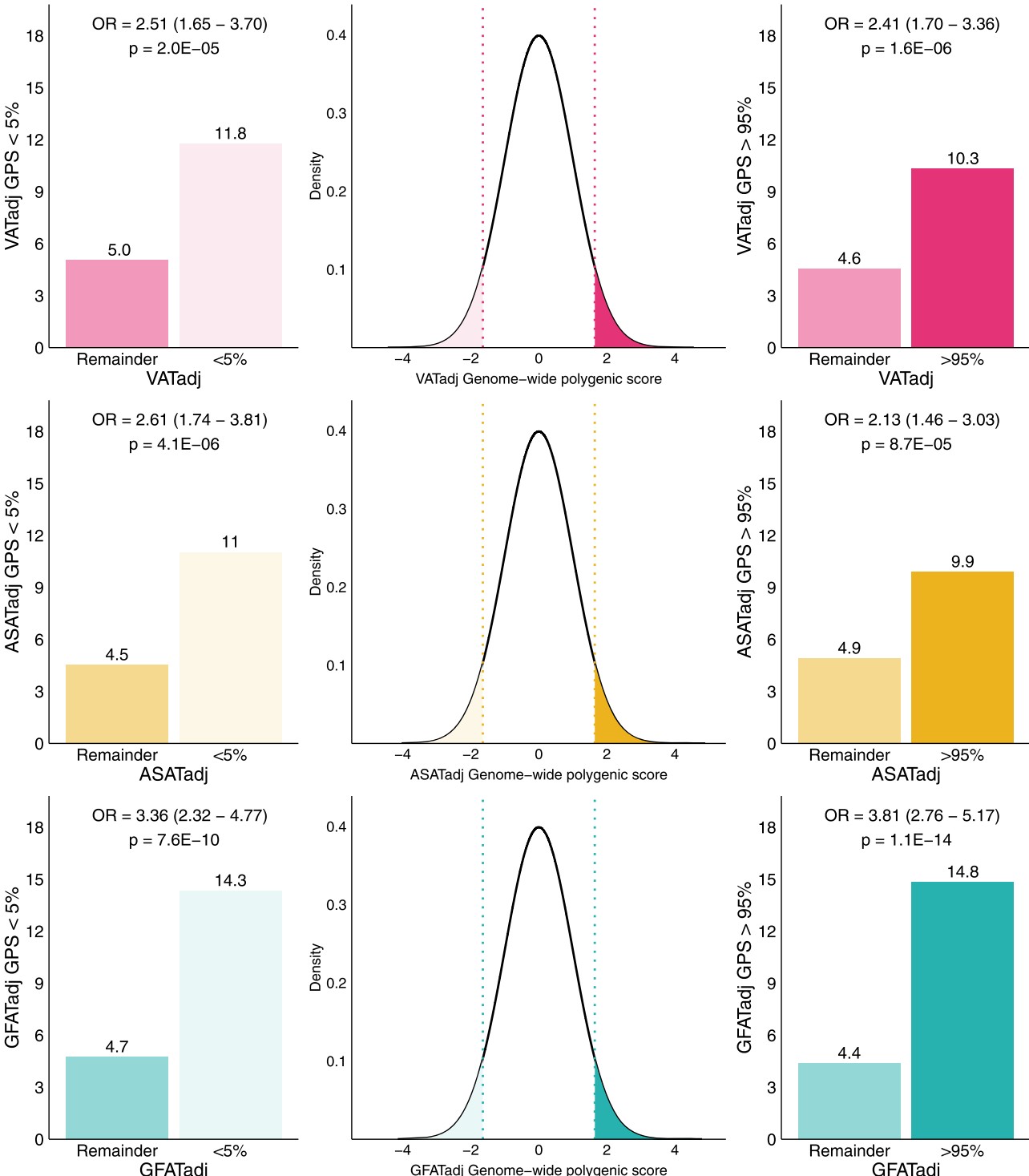

**Fig. 6 Enrichment of VATadj, ASATadj, and GFATadj genome-wide polygenic scores in tails of the distribution.** For each fat depot "adj" trait, a polygenic score was trained using LDpred2 on 70% of the studied cohort and a 10% validation cohort was used to determine the optimal set of hyperparameters. Results in this figure correspond to the 20% imaged and testing set ($N = 7795$). Supplementary Fig. 18 shows the full distribution of each polygenic score in each tail of VATadj, ASATadj, and GFATadj.

participants. An additional 10% of participants was used as training data to select optimal LDpred2 hyperparameters and the remaining 20% of participants were held out for testing. In the test set, VATadj, ASATadj, and GFATadj polygenic scores explained 5.8%, 3.6%, and 7.0% of the corresponding trait variance, respectively (Supplementary Data 18 and 19). Participants at the tails of the distribution for any of the three local adiposity

traits were enriched in extreme polygenic scores—for example, participants in the top 5% of the GFATadj distribution were nearly four times as likely to have a GFATadj polygenic score in the top 5% of the distribution (14.8% vs. 4.4%; OR = 3.81; 95% CI: 2.76–5.17) (Fig. 6 and Supplementary Fig. 18). Conversely, individuals with less than the 5th percentile of GFATadj were over three times as likely to have a GFATadj polygenic score less

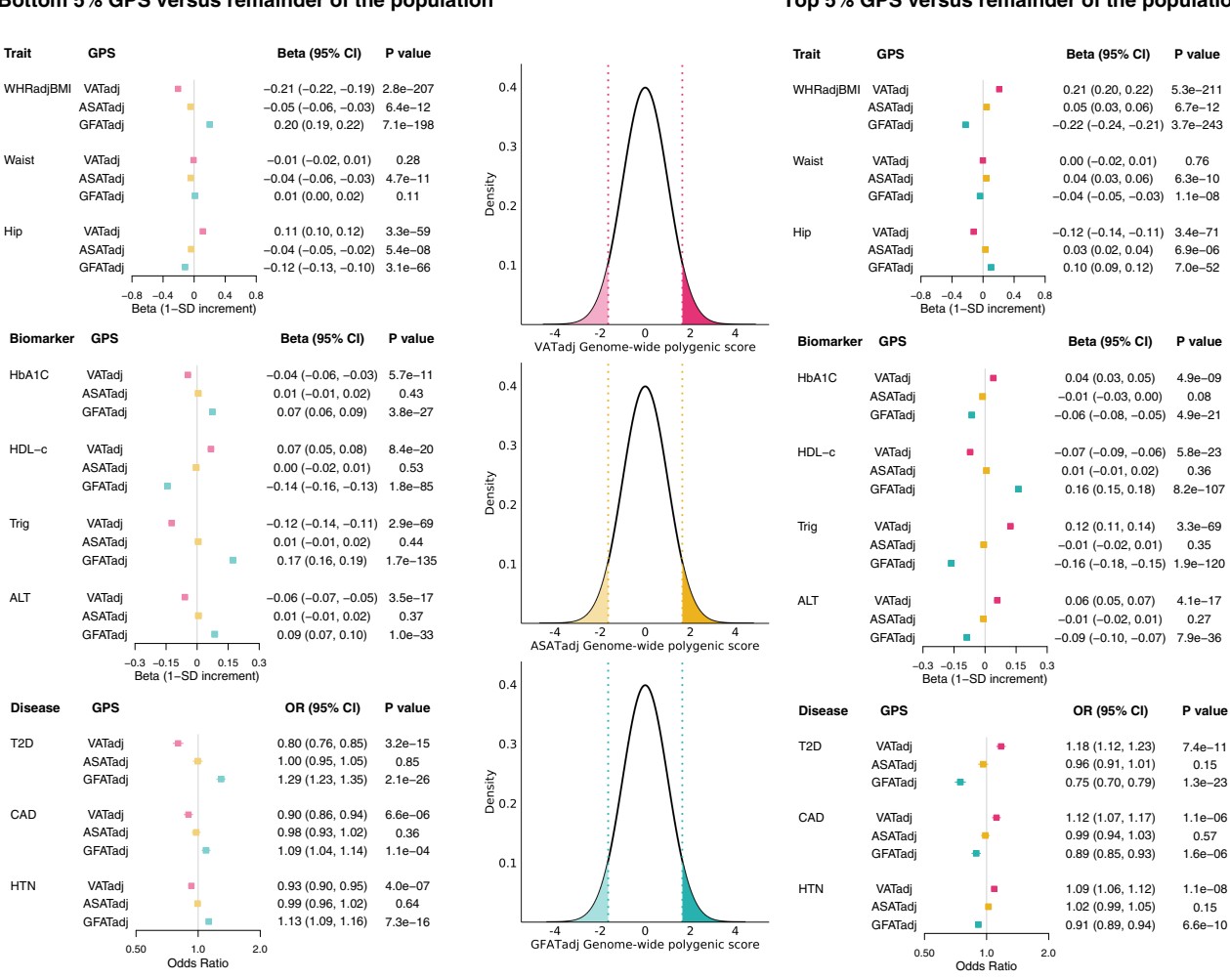

**Fig. 7 Effects of VATadj, ASATadj, and GFATadj polygenic scores on metabolically relevant biomarkers and diseases.** The central density plots indicate the distributions of VATadj, ASATadj, and GFATadj polygenic scores in genotyped individuals of the UK Biobank who were not imaged (*N* = 447,486). The dotted lines and shaded regions correspond to individuals in the top 5% and bottom 5% of the polygenic score. Forest plots to the right correspond to effect sizes of an indicator variable for being in the top 5% of the polygenic score (with identical color-coding to the density plots), while forest plots to the left correspond to effect sizes of an indicator variable for being in the bottom 5% of the polygenic score. Each polygenic score was residualized against the first ten principal components of genetic ancestry prior to being discretized, and each regression was adjusted for age at imaging, sex, and the first ten principal components of genetic ancestry. HbA1C hemoglobin A1C, HDL-c HDL-cholesterol, Trig triglycerides, ALT alanine aminotransferase, T2D prevalent type 2 diabetes (at time of imaging), CAD prevalent coronary artery disease, HTN prevalent hypertension. Corresponding data are found in Supplementary Data 20.

than the 5th percentile (14.3% vs. 4.7%; OR = 3.36; 95% CI: 2.32–4.77). These findings suggest that polygenic inheritance plays an important role in fat distribution, and that polygenic scores could feasibly be used to enrich cohorts for individuals with extreme imaging phenotypes.

We next tested the relationship between VATadj, ASATadj, and GFATadj polygenic scores and biomarkers of metabolic health (hemoglobin A1C, HDL cholesterol, serum triglycerides, and alanine aminotransferase (ALT)) and disease outcomes (type 2 diabetes, hypertension, and coronary artery disease) (Fig. 7 and Supplementary Data 20).

Within an independent dataset of 447,486 individuals of the UK Biobank who were genotyped, but not imaged, individuals in the top 5% of the GFATadj polygenic score had higher HDL-cholesterol (beta: 0.16 SD; 95% CI: 0.15–0.18; $p = 8.2 \times 10^{-107}$), lower serum triglycerides (beta: −0.16 SD; 95% CI: −0.18−−0.15; $p = 1.9 \times 10^{-120}$), lower serum ALT (beta: −0.09; 95% CI:

−0.10−−0.07; $p = 7.9 \times 10^{-36}$), lower risk of type 2 diabetes (OR: 0.75; 95% CI: 0.70–0.79; $p = 1.3 \times 10^{-23}$), and lower risk of coronary artery disease (OR: 0.89; 95% CI: 0.85–0.93; $p = 1.6 \times 10^{-6}$). By contrast, those in the top 5% of the VATadj polygenic score tended to have increased risk of these disease outcomes with odds ratios for type 2 diabetes, coronary artery disease, and hypertension of 1.18, 1.12, and 1.09, respectively.

We aimed to externally validate associations with VATadj, ASATadj, and GFATadj polygenic scores in 7888 White participants of the Atherosclerosis Risk in Communities (ARIC) study[72]. Each polygenic score was associated with HDL-cholesterol, triglycerides, and type 2 diabetes in ARIC. Results were broadly consistent with the UK Biobank with the strongest associations observed with the GFATadj polygenic score— individuals in the top 10% of the GFATadj polygenic score had higher HDL-cholesterol (beta: 0.14 SD, 95% CI: 0.07–0.22, $p = 1.5 \times 10^{-4}$), lower serum triglycerides (beta: −0.16 SD; 95%

CI: −0.23−−0.08, $p = 3.2 \times 10^{-5}$), and lower risk of prevalent type 2 diabetes (OR: 0.57; 95% CI: 0.41–0.78, $p = 5.5 \times 10^{-4}$) (Supplementary Data 21).

## Discussion

In this study, we investigated the inherited basis of body fat distribution using VAT, ASAT, and GFAT volumes quantified from body MRI in up to 38,965 individuals. Local adiposity traits derived from these fat depots had a significant inherited component, enabling identification of 250 unique loci across all traits. The increased precision afforded by image-derived quantification confirmed and extended prior work indicating significant sex-dimorphism, refined depot-specific associations for loci previously identified for WHRadjBMI, and led to the discovery of newly-associated loci, including a missense variant in *SERPINA1* that predisposes to a metabolically healthier fat distribution. Polygenic scores for local adiposity traits were highly enriched among those with "lipodystrophy-like" fat distributions and were associated with cardiometabolic traits in a depot-specific fashion. These results have at least four implications.

First, traits aiming to quantify variation in body habitus—even when they are image-derived measurements of specific fat depot volumes as in this study—tend to be highly observationally and genetically correlated with one another and with BMI. GWAS of raw VAT, ASAT, and GFAT volumes each identified a well-known intronic *FTO* variant—characteristic of BMI—as a top signal, and cell-enrichment analyses of each unadjusted fat depot displayed a pattern of CNS cell-enrichment, consistent with the signal for BMI[64]. By contrast, fat depot volumes adjusted-for-BMI-and-height and fat depot ratios—traits that capture local adiposity—were more heritable than measures of overall adiposity, revealed depot-specific genetic architecture, and displayed a pattern of adipose tissue cell-enrichment. As large cohorts with body imaging become more prominent, careful consideration of this correlation structure is warranted to enable interpretation of genetic association results. For example, a measurement of VAT predicted from a model using primarily anthropometric traits was very highly genetically correlated with BMI ($r_g = 0.93$), suggesting that the resultant genetic associations may predominantly reflect a component of VAT that is complementary to VATadj ($r_g$ with BMI = −0.16) in this study[29]. Additional investigation of how best to utilize composite phenotypes that jointly represent several correlated adiposity traits may prove useful[73,74].

Second, GFAT is highly heritable (GFATadj $h^2 = 0.41$)—particularly in females (GFATadj $h^2 = 0.52$)—with a genetic architecture that is distinct from VAT and ASAT when adjusted for overall adiposity. Most prior genetic studies of imaging-derived adiposity traits to date have been limited to VAT and ASAT—in this study, only 13 of 54 genome-wide significant loci for GFATadj overlapped with either VATadj or ASATadj[26–28]. Individuals with a GFATadj polygenic score in the bottom 5% were enriched for adverse cardiometabolic biomarker profiles and increased risk of type 2 diabetes and coronary artery disease. These observations lend further support to the hypothesis that a primary insult in a metabolically unhealthy fat distribution is the inability of the gluteofemoral fat depot to adequately expand[4,75]. Additional study of GFAT depots—or related measures such as gynoid fat from DEXA scans—in future biobank-scale studies is warranted to determine the consistency of these genetic associations across diverse age and ancestry groups.

Third, this study extends prior work suggesting that common genetic variation—as captured by a polygenic score—contributes to extreme fat distribution phenotypes[10,19,20,70]. While several of the familial partial lipodystrophies (FPLD) are known to be caused by monogenic variation in genes like *LMNA* and *PPARG*, FPLD type 1 has not been linked to a single mutation, leading some to suggest that this disease may be polygenic in nature[10]. Lotta et al. provided evidence for this by demonstrating that individuals with FPLD1 had a higher burden of a 53-SNP insulin resistance polygenic score compared to the general population[19]. In this study, individuals who harbor lower than average GFA-Tadj or ASATadj and/or higher than average VATadj tended to manifest a mild lipodystrophy-like phenotype. We demonstrate that individuals at the extremes of these local adiposity traits are enriched in extreme polygenic scores suggesting that polygenic scores may be helpful in identifying this subgroup of individuals for future focused investigations. For example, growth hormone releasing hormone analogs—such as tesamorelin—have previously been shown to lead to a selective reduction of VAT in patients with obesity or HIV-associated lipodystrophy[76,77]. Whether a local adiposity polygenic score—perhaps in combination with emerging imaging tools for identifying lipodystrophies—could identify a subset of individuals with obesity and polygenic lipodystrophy who may benefit from these fat redistribution agents in addition to traditional obesity therapy is an area for future investigation[78].

Fourth, these results lay the scientific foundation for variant-to-function studies to link fat distribution-associated genetic risk loci to effector genes and mechanisms of action in depot-specific adipocyte model systems[79]. Such targeted perturbation studies in subcutaneous and visceral adipocyte cell lines may reveal key biological pathways driving fat distribution, and may generate therapeutic hypotheses for adverse fat distribution-related traits[19,80].

In conclusion, we carried out genetic association studies of local adiposity traits in a large cohort of individuals with MRI imaging. Our work characterizes the depot-specific genetic architecture of visceral, abdominal subcutaneous, and gluteofemoral adipose tissue, and extends efforts to define and identify individuals with polygenic lipodystrophy.

## Methods

**Study population.** The UK Biobank is an observational study that enrolled over 500,000 individuals between the ages of 40 and 69 years between 2006 and 2010, of whom 43,521 underwent MRI imaging between 2014 and 2020[81,82]. Our group previously estimated VAT, ASAT, and GFAT volumes in 40,032 individuals of the imaged cohort after excluding 3489 (8.0%) scans based on technical problems or artifacts[5]. A subset of 39,076 individuals with genotype array data available was studied here. Compared to non-imaged individuals of the UK Biobank at enrollment, imaged individuals were younger (mean age 56 years vs. 57 years), less likely to be female (51% vs. 55%), and more likely to be of white British ancestry (87% vs. 84%) (Supplementary Data 2). Individuals were not excluded on the basis of ancestry. This analysis of data from the UK Biobank was approved by the Mass General Brigham institutional review board and was performed under UK Biobank application #7089.

**Deriving local adiposity traits.** The focus of this study was to investigate the genetic architecture of fat distribution independent of the overall size of an individual. Two sets of traits were derived for this purpose: "adj" traits and fat depot ratios. "adj" traits represent residuals of the fat depot in question in sex-specific linear regressions against age, age squared, BMI, and height. We provide justification in the Supplementary Methods for adjusting for both BMI and height as opposed to only BMI. In brief, adjusting only for BMI introduces a significant genetic correlation of each adj trait with height (most pronounced with ASAT and GFAT). Several prior studies have suggested that adjusting for heritable covariates can lead to spurious genetic associations due to collider bias[83,84]. We investigated the extent to which VATadj, ASATadj, and GFATadj loci may be driven by collider bias with BMI or height and found little evidence for collider bias making a significant contribution to these results (Supplementary Methods and Supplementary Data 22).

**Genotyping, imputation, and QC.** Genotyping in the UK Biobank was done with two custom genotyping arrays: UK BiLEVE and Axiom[85]. Imputation was done using the UK10K and 1000 Genomes Phase 3 reference panels[86,87]. Prior to analysis, genotyped SNPs were filtered based on the following criteria, only including variants if: (1) MAF ≥ 1%, (2) Hardy-Weinberg equilibrium (HWE) $p > 1 \times 10^{-15}$,

(3) genotyping rate ≥ 99%, and (4) LD pruning using $R^2$ threshold of 0.9 with window size of 1000 markers and step size of 100 marker[88,89]. This process resulted in 433,616 SNPs available for genetic relationship matrix (GRM) construction. Imputed SNPs with MAF < 0.005 or imputation quality (INFO) score <0.3 were excluded. Note that the MAF filter was applied to the UK Biobank imputed file prior to subsetting to the imaged substudy. These criteria resulted in a total of 11,485,690 imputed variants available for analysis.

Participant were excluded from analysis if they met any of the following criteria: (1) mismatch between self-reported sex and sex chromosome count, (2) sex chromosome aneuploidy, (3) genotyping call rate <0.95, or (4) were outliers for heterozygosity. Up to 38,965 participants were available for analysis (37,641 for adj traits because these individuals also had to have BMI and height available).

**Common variant association studies.** Nine traits were analyzed (VAT, ASAT, GFAT, VATadj, ASATadj, GFATadj, VAT/ASAT, VAT/GFAT, and ASAT/GFAT) in three contexts (sex-combined, male only, female only), leading to 27 analyses in total. SNP-heritability was estimated using BOLT-REML v2.3.4[90,91]. Genetic correlations between traits were estimated using cross-trait LD-score regression (*ldsc* v1.0.1) using default settings[33,34].

Prior to conducting GWAS, each trait was inverse-normal transformed. Each analysis was adjusted for age at the time of MRI, age squared, sex (except in sex-stratified analyses), the first ten principal components of genetic ancestry, genotyping array, and MRI imaging center. BOLT-LMM v2.3.4 was used to carry out GWAS accounting for cryptic population structure and sample relatedness[90,91]. After the QC protocol detailed above, 433,616 SNPs were available for GRM construction. A threshold of $p < 5 \times 10^{-8}$ was used to denote genome-wide significance, while a threshold of $p < 5 \times 10^{-8}/27 = 1.9 \times 10^{-9}$ was used to denote study-wide significance.

Lead SNPs were prioritized with LD clumping. LD clumping was done with the -clump function in PLINK to isolate independent signals for each GWAS. The parameters were as follows: -clump-p1 5E−08, -clump-p2 5E−06, -clump-r2 0.1, -clump-kb 1000, which can be interpreted as follows: variants with $p < 5E-08$ are chosen starting with the lowest $p$ value, and for each variant chosen, all other variants with $p < 5E-06$ within a 1000 kb region and $r^2 > 0.1$ with the index variant are assigned to that index variant. This process is repeated until all variants with $p < 5E-08$ are assigned an LD clump. An LD reference panel for this task was constructed using a random sample of 3000 individuals from the studied.

The extent of genomic inflation vs. polygenicity was assessed by computing the LD-score regression intercept (*ldsc* v1.0.1) using default settings[33].

A lead SNP was defined as newly-identified if it was not in LD ($R^2 < 0.1$) with any SNP in the GWAS catalog (downloaded June 08, 2021) with genome-wide significant association ($p < 5 \times 10^{-8}$) with any "DISEASE/TRAIT" containing the following characters: (1) "body mass", (2) "BMI", (3) "adipos", (4) "fat", (5) "waist", (6) "hip circ", or (7) "whr". These characters captured key anthropometric traits of interest (e.g., BMI, waist circumference, hip circumference, waist-to-hip ratio) as well as other related traits of interest (e.g., VAT, predicted VAT, fat impedance measures).

**Clustering to classify gluteofemoral adiposity signals.** Clustering analysis was performed for GFATadj and GFAT association signals. We started with all 250 lead SNPs significantly associated with any of the nine adiposity traits and extracted those associated with the primary trait (e.g., GFATadj) with nominal significance ($p < 0.05$) for each analysis. To ensure that only independent signals were used for the clustering, variants were LD-pruned using a LD threshold of $r^2 = 0.1$. When two SNPs were found to be in LD above this threshold, the variant with the lower $p$ value was retained.

Summary statistics were gathered from GWAS performed in the UK Biobank for 32 cardiometabolic traits (Supplementary Data 6). For each first GWAS, the regression coefficient betas was divided by the SE to obtain standardized effect sizes. These standardized effects were further scaled by dividing by the square root of the variant's sample size for the given trait GWAS and then multiplying by the square root of the median sample size of all GWAS. Since all summary statistics were sourced from UK Biobank, this additional scaling had a negligible effect.

The clustering traits were then filtered to retain those relevant to the analysis by removing any that were not associated with at least one variant at a Bonferroni $p$ value threshold (0.05/number of SNPs). When two traits had highly correlated Z-scores ($|r| > 0.85$), the trait with the lower minimum $p$ value was kept and the other removed. The remaining standardized effect sizes made up the variant-trait association matrix, Z (N variants by M traits).

In order to satisfy the non-negative requirement of Bayesian non-negative matrix factorization (bNMF), each column was split into two arrays: one with the positive Z-scores and the other with the absolute value of the negative Z-scores. This means that the final association matrix, X, contained N variants by 2M traits.

The bNMF clustering was performed as previously described[20]. The procedure attempts to approximate the association matrix by factorizing X into two matrices, W (2M by K) and HT (N by K), with an optimal rank K. bNMF is designed to suggest an optimal K best explaining X at the balance between an error measure, $||X - WH||^2$, and a penalty for model complexity derived from a non-negative half-normal prior for W and H. In addition, bNMF exploits an automatic relevance determination technique to iteratively regress out irrelevant components in

explaining the observed data X. The exact objective function optimized by bNMF is a posterior, which has two opposing contributions from the likelihood (Frobenius norm) and the regularization penalty (L2-norm of W and H coupled by the relevance weights). For all analyses, bNMF was run with 100 iterations for each. All analyses converged in ≥92% of iterations to their given K solution. Code used in the bNMF clustering is available on GitHub: https://github.com/kwesterman/bnmf-clustering.

**Identification of sex-dimorphic signals.** Genetic correlations between sexes for each of the adiposity traits were computed using cross-trait LD-score regression as described above.

Using sex-specific GWAS summary statistics for each of the six local adiposity traits (VATadj, ASATadj, GFATadj, VAT/ASAT, VAT/GFAT, ASAT/GFAT), we tested each of the 220 genetic loci that were genome-wide significant for any of the six local adiposity traits in either sex-combined or sex-stratified analyses for sex dimorphism by computing the $t$-statistic:

$$t = \frac{\text{beta(male)} - \text{beta(female)}}{\sqrt{\text{se(male)}^2 + \text{se(female)}^2 - 2 \times r \times \text{se(male)} \times \text{se(female)}}} \quad (1)$$

where beta is the effect size for an adiposity trait in sex-stratified GWAS, se is the standard error, and $r$ is the genome-wide Spearman rank correlation coefficient between males and females. The $t$-statistic and associated $p$ value ($p_{\text{diff}}$) were computed using the EasyStrata software[92]. Given that 220 independent loci were tested, a significance threshold of $p_{\text{diff}} < 0.05/220 = 2.3 \times 10^{-4}$ was used.

**WHRadjBMI loci lookups.** A recent meta-analysis for the WHRadjBMI trait across 694,649 individuals revealed 346 unique associated loci[12]. Of these 346 loci, the primary signals for 345 loci were among the imputed variants available for analysis in this study. We plotted the effect sizes for VATadj, ASATadj, and GFATadj for each of these 345 loci and further quantified the frequency of "WHRadjBMI-discordance" defined as either (1) WHRadjBMI and VATadj effects going in opposite directions, (2) WHRadjBMI and ASATadj effects going in opposite directions, or (3) WHRadjBMI and GFATadj effects going in the same direction. For each adiposity trait in the "WHRadjBMI-discordance" analysis, we excluded loci for which the effect size beta was smaller than the SE to avoid inflating the fraction of "WHRadjBMI-discordant" loci.

**External validation with prior meta-analysis.** External validation for 76 genome-wide significant SNP-trait associations with VAT, ASAT, VATadj, and VAT/ASAT ratio was pursued using summary statistics downloaded from the GWAS catalog of a multiethnic genome-wide meta-analysis of ectopic fat depots in up to 2.6 million SNPs in up to 18,332 individuals[27,35]. Alleles were aligned and the $z$-score for each SNP from the previous study were compared with the effect sizes in the current study to determine concordance.

**Transcriptome-wide association study.** For each of the six local adiposity traits (VATadj, ASATadj, GFATadj, VAT/ASAT, VAT/GFAT, ASAT/GFAT), we performed a TWAS to prioritize genes on the basis of imputed cis-regulated gene expression using FUSION with default settings[60,93,94]. Pre-computed gene expression weights from GTEx v7 were used as downloaded from the FUSION website (http://gusevlab.org/projects/fusion/)[60]. Reference weights for visceral adipose tissue were used for VATadj, while those for subcutaneous adipose tissue were used for ASATadj, GFATadj, and ASAT/GFAT ratio. Weights from both visceral and subcutaneous adipose tissue were used for VAT/ASAT and VAT/GFAT ratios.

**Cell- and tissue-specific enrichment.** We used stratified LD-score regression to identify cell types that are most relevant for each of the nine adiposity traits (VAT, ASAT, GFAT, VATadj, ASATadj, GFATadj, VAT/ASAT, VAT/GFAT, and ASAT/GFAT) and BMI[64]. We carried out this analysis using *ldsc* v1.0.1 with default settings and using two gene expression datasets that are described in the manuscript outlining stratified LD-score regression[64]: GTEx[95] and the "Franke lab"[96,97] dataset.

**Sequencing and sample quality control for rare-variant association study.** We conducted rare-variant association studies using data from the 200,643 exomes released by the UK Biobank[98]. Whole-exome sequencing was performed by the Regeneron Genetics Center using an updated Functional Equivalence protocol that retains original quality scores in the CRAM files (referred to as the OQFE protocol) as previously described[98]. The DTxGen Exome Research Panel v1.0 including supplemental probes was used for exome capture for this dataset (https://biobank.ctsu.ox.ac.uk/showcase/label.cgi?id=170). In total, 19,396 genes in the targets of 38Mbp were covered. In total, 75 × 75 bp paired-end reads were sequenced on the Illumina NovaSeq 6000 platform. For each sample in the targeted region, more than 95.2% of sites were covered by more than 20 reads. We downloaded the pVCF file provided by the UK Biobank, and then applied additional genotype call, variant, and sample quality control[99].

The individual genotype call was set as missing if reads depth (DP) ≤ 10 or DP ≥ 200, if homozygous reference allele with genotype quality (GQ) ≤ 20 or the ratio of alt allele reads over all of the covered reads >0.1, if heterozygous with the ratio of alt allele reads over all of the covered reads <0.2 or Phred-scaled likelihood (PL) of the reference allele <20, or if homozygous alternate with the ratio of alt allele reads over all of the covered reads <0.9 or PL of reference allele <20. The variant quality control was performed using the following exclusion criteria:

- Variants in low-complexity regions of the genome that preclude accurate read alignment as previously defined[100].
- Variants in segmental duplication region of the genome[100,101].
- Hardy-Weinberg disequilibrium (HWE) $p$ value $<1 \times 10^{-15}$.
- Variant call rate <90%.
- Monomorphic sites after the above genotype call quality control.

After the above genotype call and variant QC, we selected a subset of high-quality variants for inferring the genetic kinship matrix and genetic sex used for sample QC. We selected independent autosome variants by MAF > 0.1%, missingness <1%, and HWE $p > 10^{-6}$. We further pruned the variants using PLINK2 software[102] with a window size of 200, step size 100, and $R^2 = 0.1$ and removed indels and strand ambiguous SNPs. Based on these variants, we used KING (version 2.2.5)[103] to infer the genetic kinship matrix. We further selected X-chromosomal variants, not within the pseudo-autosomal regions, based on the sample variant QC criteria as for the autosome variants and did the same variant pruning procedure. We then inferred the genetic sex based on the $F$ statistics by PLINK2 software, $F > 0.8$ was set to male, while samples with $F < 0.5$ were set to female. Eighty samples were removed because of the discordance of genetic sex with self-reported sex. We further removed samples if:

- The ratio of heterozygote/homozygote beyond 8 standard deviations ($N = 100$ samples removed).
- The ratio of the number of SNVs/indels beyond 8 standard deviations ($N = 1$ samples removed).
- The number of singletons was beyond 8 standard deviations ($N = 111$ samples removed).
- Genotype call rate <90% ($N = 1$ sample removed).
- Withdrawal of informed consent ($N = 13$ samples removed).

We further randomly removed one sample if a pair of samples had second-degree relative or closer kinship, defined as kinship coefficient >0.088474 ($N = 1563$ samples removed). Of all the above QC passed samples, 19,255 samples out of the 40,032 having image-derived traits were used in the downstream rare variant burden test. We converted the genetic coordinates from GRCh38 to GRCh37 using CrossMap software (version: v0.3.3)[104].

**Approach to variant annotation and weighting**. To identify rare (MAF <0.1%) high-confidence predicted inactivating variants, we applied the previously validated Loss-Of-Function Transcript Effect Estimator (LOFTEE) algorithm implemented within the Ensembl Variant Effect Predictor (VEP) software program as a plugin, VEP version 96.0[105,106]. The LOFTEE algorithm identifies stop-gain, splice-site disrupting, and frameshift variants. The algorithm includes a series of flags for each variant class that collectively represent "low-confidence" inactivating variants. In this study, we studied only variants that were "high-confidence" inactivating variants without any flag values. This aggregation strategy will be referred to hereafter as putative loss-of-function ("pLoF").

To identify rare (MAF <0.1%) predicted damaging missense variants, we included variants predicted to be damaging by all of five computational prediction algorithms[107–109]. In brief, predictions were retrieved from the dbNSFP database[110], version 2.9.3, with the most severe prediction across multiple transcripts used. We focused on five prediction algorithms: SIFT[111] (including variants annotated as damaging), PolyPhen2-HDIV and PolyPhen2-HVAR[112] (including variants annotated as possibly or probably damaging), LRT[113] (including variants annotated as deleterious), and MutationTaster[114] (including variants annotated as disease-causing-automatic or disease-causing). Within the association testing framework, this class of variants was given a gene-specific weight based on the relative cumulative frequency of these predicted damaging missense variants as compared to the cumulative frequency of high-confidence predicted inactivating variants identified by LOFTEE algorithm using a previously recommended approach:[115,116] given the cumulative allele frequency of all of the LOFTEE high-confidence rare variants of a gene ($G$) as $f_L$, the cumulative allele frequency of all of the predicted damaging missense variants as $f_M$, the weight for the missense variants was estimated as the quantity in Eq. (2) and capped at 1.0:

$$\left( \frac{f_L (1 - f_L)}{f_M (1 - f_M)} \right)^{0.5} \quad (2)$$

For genes without LOFTEE high-confidence rare variants, the weight for missense variants is 1.0. This aggregation strategy will be referred to hereafter as putative loss-of-function plus missense ("pLoF + missense").

**Statistical analysis**. We tested the association between the aggregated rare variant score (the weighted sum of the qualified variant of each gene) and each inverse normal transformed phenotype using a multivariable regression model in sex-combined and sex-stratified models. Analyses were restricted to genes that had at least ten variant carriers in the analyzed cohort. An individual's gene-specific score was computed according to the weighting strategy described above and capped at one. The covariates were the same as the common variant association test. Given the filter of ten variant carriers, sex-combined analyses tested 12,020 genes and so a gene was recognized as exome-wide significant if the gene's p value was smaller than the Bonferroni-corrected $p$ value threshold of $0.05/12{,}020 = 4.2 \times 10^{-6}$.

**Polygenic score**. We used the LDpred2 algorithm[71] to derive genome-wide polygenic scores for each trait. We randomly selected 350,000 White British ancestry individuals from the UK Biobank to use as the LD reference panel[85], and used HapMap3 variants with MAF > 0.5% in the LD reference panel to compute the LD correlation matrix. For each trait, we partitioned the samples into three independent portions: 70% to run the GWAS for making the summary statistics, 10% to select the optimal hyperparameters, and 20% to test performance. We randomly removed one sample in a pair if the pair had a genetic relationship closer than a second-degree genetic relationship in the last two partitions of samples, and checked the pairwise relationship across the whole dataset. For the hyperparameters of the LDpred2 algorithm, we grid searched three parameters: (1) 0.7, 1, and 1.4 times of genome-wide heritability estimation, (2) whether or not to use a sparse LD correlation matrix, and (3) 17 different estimates of the proportion of causal variants selecting from $[0.18, 0.32, 0.56, 1] \times 10^{[0, -1, -2, -3]}$ and 0.0001. In total, we tested $3 \times 2 \times 17 = 102$ grid points.

For all downstream analyses, each polygenic score was residualized against the first ten principal components of genetic ancestry prior to regression with the dependent variable of interest, and each regression was adjusted for age at the time of imaging, sex, and the first ten principal components of genetic ancestry.

**Polygenic score external validation in ARIC**. The ARIC study is a prospective cohort study that—beginning in 1987—enrolled white and black participants between the ages of 45 and 64 years[72]. Genotype and clinical data were retrieved from the National Center for Biotechnology Information dbGAP server (accession number phg000035.v1). VATadj, ASATadj, and GFATadj polygenic scores were computed using identical LDpred2 weights and the optimal hyperparameter set for UK Biobank analyses. Circulating biomarkers and clinical risk factor ascertainment was performed at time of enrollment as previously described[72].

**Reporting summary**. Further information on research design is available in the Nature Research Reporting Summary linked to this article.

## Data availability
This research has been conducted using the UK Biobank Resource under Application Number #7089. The raw UK Biobank data are made available to researchers from universities and other research institutions with genuine research inquiries, following IRB and UK Biobank approval. The GWAS summary statistics and polygenic score weights generated in this study will be made available for download from the Cardiovascular Disease Knowledge Portal under the "Downloads" tab at https://cvd. hugeamp.org/ at the time of publication. All other relevant results are available in the Supplementary Data.

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

## Acknowledgements

This work was supported by the Sarnoff Cardiovascular Research Foundation Fellowship (to S.A.), student scholarships from the Dutch Heart Foundation and the Amsterdams Studentenfonds (S.J.J.), grants 1K08HG010155 and 1U01HG011719 (to A.V.K.) from the National Human Genome Research Institute, a grant from the National Institute of Diabetes and Digestive and Kidney Diseases K23DK114551 (to M.S.U.), a Hassenfeld Scholar Award from Massachusetts General Hospital (to A.V.K.), a Merkin Institute Fellowship from the Broad Institute of MIT and Harvard (to A.V.K.), and a sponsored research agreement from IBM Research to the Broad Institute of MIT and Harvard (to P.T.E., A.P., P.B., and A.V.K.).

## Author contributions

S.A., M.W., and A.V.K. conceived and designed the study. S.A., M.W., M.D.R.K., J.S., K.S., and N.D. acquired, analyzed, and interpreted the data. S.A., M.W., and A.V.K. drafted the manuscript. S.A., M.W., M.D.R.K., J.S., H.D., N.D., S.H.C., S.J.J., P.T.E., A.P.,

K.N., M.C., P.B., M.S.U., and A.V.K. critically revised the manuscript for important intellectual content.

## Competing interests

S.A. has served as a scientific consultant to Third Rock Ventures. M.D.R.K., A.P., and P.B. are supported by grants from Bayer AG applying machine learning in cardiovascular disease. P.T.E. receives sponsored research support from Bayer AG and IBM and has consulted for Bayer AG, Novartis, MyoKardia and Quest Diagnostics. A.P. is also employed as a Venture Partner at GV and consulted for Novartis; and has received funding from Intel, Verily and MSFT. M.C. holds equity in Waypoint Bio and is a member of the Nestle Scientific Advisory Board. K.N. is an employee of IBM Research. P.B. serves as a consultant for Novartis. A.V.K. is an employee and holds equity in Verve Therapeutics; has served as a scientific advisor to Amgen, Maze Therapeutics, Navitor Pharmaceuticals, Sarepta Therapeutics, Novartis, Silence Therapeutics, Korro Bio, Veritas International, Color Health, Third Rock Ventures, Illumina, Foresite Labs, and Columbia University (NIH); received speaking fees from Illumina, MedGenome, Amgen, and the Novartis Institute for Biomedical Research; and received a sponsored research agreement from IBM Research. The remaining authors declare no competing interests.
