## [Peer Review File · Nature Communications]

Inherited basis of visceral, abdominal subcutaneous and gluteofemoral fat depotsEditorial Note: This manuscript has been previously reviewed at another journal that is not operating a transparent peer review scheme. This document only contains reviewer comments and rebuttal letters for versions considered at *Nature Communications*. Mentions of the other journal have been redacted.

REVIEWER COMMENTS

Reviewer #1 (Remarks to the Author):

I have no further comments. The authors have thoroughly addressed the comments I had with the [REDACTED] version.

Reviewer #2 (Remarks to the Author):

The revised manuscript is much improved and the authors have directly addressed all my concerns. The clarified description and naming of the adjusted traits is a major improvement, and their reduced association with body size is described well.

Other comments:

- Figure 2 would warrant the inclusion of height given the importance of lack of genetic correlation with height to the results.
- I agree with the other reviewer that the shared enrichment of individuals in the extremes of the phenotypic/genotypic distributions (figure 6) is an expected result - one that might not warrant being included in the abstract. As the reviewer points out they do not illustrate the potential misclassification because they do not show what % of individuals with high GPS are in the lowest part of the phenotypic distribution. Perhaps boxplots of phenotypic distribution at all deciles of the GPS would better illustrate the data and presence of outliers. I think the associations of these GPS with the other cardiometabolic traits might be more informative to readers.

Reviewer #3 (Remarks to the Author):

- Response to “A total of 27 GWASs were performed with 11 million variants. The authors state that the Bonferroni adjusted p-value is 1.9×10^{-9} a p-value threshold of 5×10^{-9} is assumed for a single GWAS, or 2×10^{-10} for 27 GWASs.

The authors argue that others that used imaging data used the traditional threshold, instead of the increasingly used more stringent 5×10^{-9} . However, the threshold does not depend on the phenotype, but on the number of variants tested. As GWAS with UK Biobank data screens for many more loci than much smaller studies in the past, it is warranted to account for more independent tests and thus use the $P < 5 \times 10^{-9}$ threshold.

The authors have completely ignored the equally important comment that the significance threshold needs to account for the 27 traits that have been analyzed.

Taken together, a more significant threshold is needed.

- Response to “It seems that the so-called “BMI-adjusted” analyses are not only adjusted for BMI, but also for height. ... However, no mention of the additional adjustment is made until the methods section, where no justification for this additional adjustment is given.” And “Related to the previous topic, an important concern when traits are adjusted for other correlated traits (such as BMI and height) is the introduction of collider bias. I would strongly recommend testing which variants have been impacted by collider bias and account for this in follow up analyses.”

Despite their lengthy response, the collider bias was clearly not properly addressed in the paper. The authors need to perform a formal, quantitative analysis to determine whether or not there is collider bias. Examples can be found in e.g. Pulit et al HMG 2019.

Also, one can keep adjusting for other traits. However, traits lose their purpose and cannot be interpreted anymore – as is the case in the current paper.

And as mentioned before, “As BMI is already a function of height, i.e. $\text{weight}/\text{height}^2$, and BMI and height tend to be correlated, this additional adjustment for height results in an odd phenotype that becomes hard to interpret.”

- Response to “The authors use PGSs to quantify/visualize the importance (or not) of the genetic contribution to individuals with extreme values. While a 3.8 fold greater likelihood seems impressive, ... That way, readers can interpret themselves how close (or far) we are from using genetics in predicting complex traits.”

I must have missed the answer to my comment in your response. What is the misclassification. Or, what is your answer to this question “what percentage of those with a high GFATadjBMI values (or others) do not have a high PGS (e.g. in top 5%).” ?

You did not provide any explanation with the figures, but the way I read them is that of all the individuals with a high PGS, 10.5% has indeed a high VATadjBMI (and height), whereas 89.5% does not ?

Thus, there is an enormous misclassification when using the PGS to identify individuals at high risk. This needs to be made very explicit to provide a fair message to the readers. Currently, there is no mention of the misclassification.

Response to Referees' Comments for *NCOMMS-21-45422-T*
"Inherited basis of visceral, abdominal subcutaneous, and gluteofemoral fat depots"

Reviewer #1 (Remarks to the Author):

I have no further comments. The authors have thoroughly addressed the comments I had with the [REDACTED] version.

Authors Reply: Thank you for your helpful comments.

Reviewer #2 (Remarks to the Author):

The revised manuscript is much improved and the authors have directly addressed all my concerns. The clarified description and naming of the adjusted traits is a major improvement, and their reduced association with body size is described well.

Authors Reply: Thank you for your helpful comments.

Other comments:

- Figure 2 would warrant the inclusion of height given the importance of lack of genetic correlation with height to the results.

Authors Reply: We agree that the inclusion of height in Figure 2 is warranted.

Manuscript Change(s): In response, Figure 2 was updated to include height in the revised manuscript:

- I agree with the other reviewer that the shared enrichment of individuals in the extremes of the phenotypic/genotypic distributions (figure 6) is an expected result - one that might not warrant being included in the abstract. As the reviewer points out they do not illustrate the potential misclassification because they do not show what % of individuals with high GPS are in the lowest part of the phenotypic distribution. Perhaps boxplots of phenotypic distribution at all deciles of the GPS would better illustrate the data and presence of outliers. I think the associations of these GPS with the other cardiometabolic traits might be more informative to readers.

Authors Reply: We agree that boxplots showing (1) polygenic scores in the low and high tail ends of VATadj, ASATadj, and GFATadj and (2) the distribution of each trait across deciles of the polygenic score would be a useful complement to Figure 6.

We agree with the reviewer that enrichment of high polygenic scores at the tail of the corresponding trait is an expected result for a highly heritable trait. However, given that the utility of polygenic scores is primarily in identifying those at the extremes, we feel that demonstrating the magnitude of this enrichment provides a useful snapshot for how well these polygenic scores perform. We make an effort to highlight that this enrichment does not imply that polygenic scores alone are a highly discriminatory test for each local adiposity trait by (1) including the variance explained by each polygenic score and (2) highlighting the absolute fraction of individuals in the top tail of GFATadj with a high GFATadj polygenic score.

We also agree that the associations of the GPSs derived in this study with cardiometabolic traits warrants inclusion in the abstract.

Manuscript Change(s): We have added a new **Supplementary Figure S14**.

Figure S14 Visualizing the relationship between VATadj, ASATadj, and GFATadj and their polygenic scores at the tails of the distributions

And made reference to this figure in the Results section:

“To ensure no overlap between summary statistics and tested individuals, CVAS was conducted using a randomly selected 70% of participants. An additional 10% of participants was used as training data to select optimal LDpred2 hyperparameters and the remaining 20% of participants were held out for testing. In the test set, **VATadj, ASATadj, and GFATadj polygenic scores explained 5.8%, 3.6%, and 7.0% of the corresponding trait variance**, respectively (Supplementary Table S22-S23). Participants at the tails of the distribution for any of the three local adiposity traits were enriched in extreme polygenic scores – for example, participants in the top 5% of the GFATadj distribution were nearly four times as likely to have a GFATadj polygenic score in the top 5% of the distribution (**14.8% vs. 4.4%**; OR = 3.81; 95% CI: 2.76-5.17) (Figure 6; **Supplementary Figure S14**). Conversely, individuals with less than the 5th percentile of GFATadj

were over three times as likely to have a GFATadj polygenic score less than the 5th percentile (14.3% vs. 4.7%; OR = 3.36; 95% CI: 2.32-4.77).”

We modified the Abstract to include mention of the associations between the polygenic scores and cardiometabolic traits:

“Individuals in the extreme tails of VATadj, ASATadj, and GFATadj were highly enriched for predisposing common variants, as quantified using polygenic scores. Taking GFATadj as an example, individuals with GFATadj in the top 5% were 3.8-fold (95%CI 2.8 to 5.2) more likely to have a polygenic score within the top 5% of the distribution. **A VATadj polygenic score was associated with a metabolically unhealthy profile, while a GFATadj score was associated with a metabolically healthy profile.** These results – using MRI-derived, BMI-independent measures of local adiposity – confirm fat distribution as a highly heritable trait with important implications for cardiometabolic health outcomes.”

Reviewer #3 (Remarks to the Author):

- Response to “A total of 27 GWASs were performed with 11 million variants. The authors state that the Bonferroni adjusted p-value is 1.9×10^{-9} a p-value threshold of 5×10^{-9} is assumed for a single GWAS, or 2×10^{-10} for 27 GWASs.

The authors argue that others that used imaging data used the traditional threshold, instead of the increasingly used more stringent 5×10^{-9} . However, the threshold does not depend on the phenotype, but on the number of variants tested. As GWAS with UK Biobank data screens for many more loci than much smaller studies in the past, it is warranted to account for more independent tests and thus use the $P < 5 \times 10^{-9}$ threshold.

The authors have completely ignored the equally important comment that the significance threshold needs to account for the 27 traits that have been analyzed.

Taken together, a more significant threshold is needed.

Authors Reply: We agree that appropriately correcting the p-value threshold for multiple testing is important. The prior publications we cited with similar P-value thresholds were targeted at similar audiences as the reviewer points out, but also each tested more variants than this study did (11,485,690 SNPs): the first study tested 14,134,301 SNPs, the second 13,660,711 SNPs, and the third 25,472,837 SNPs.¹⁻³ Regarding the Bonferroni correction, we agree that showing readers the number of loci that would be discovered at the new threshold would improve the paper. Notably, a Bonferroni correction is likely to be overly conservative in the setting of this paper where the 27 studies spanned only 9 unique traits that have considerably correlation structure.

Manuscript Change(s): In response, we have modified the relevant section of the Results section to note the number of loci that would have been discovered at more stringent P-value thresholds:

“Across all 27 association studies, 250 loci were associated with at least one adiposity trait at a P-value threshold of 5×10^{-8} (Supplementary Table S4). **If a more stringent genome-wide significance threshold of 5×10^{-9} had been used, we would have identified 136 loci, or 85 loci at the most conservative Bonferroni-corrected threshold of $5 \times 10^{-9} / 27 = 1.9 \times 10^{-10}$.** Of the 250 loci across all adiposity traits, 39 were newly-identified (defined as $R^2 < 0.1$ with all genome-wide significant associations with prior adiposity and relevant anthropometric traits in the GWAS catalog) (Table 1; Methods; Supplementary Tables S5).”

- Response to “It seems that the so-called “BMI-adjusted” analyses are not only adjusted for BMI, but also for height. ... However, no mention of the additional adjustment is made until the methods section, where no justification for this additional adjustment is given.” And “Related to the previous topic, an important concern when traits are adjusted for other correlated traits (such as BMI and height) is the introduction of collider bias. I would strongly recommend testing which variants have been impacted by collider bias and account for this in follow up analyses.”

Despite their lengthy response, the collider bias was clearly not properly addressed in the paper. The authors need to perform a formal, quantitative analysis to determine whether or not there is collider bias. Examples can be found in e.g. Pulit et al HMG 2019.

Also, one can keep adjusting for other traits. However, traits lose their purpose and cannot be interpreted anymore – as is the case in the current paper.

And as mentioned before, “As BMI is already a function of height, i.e. $\text{weight}/\text{height}^2$, and BMI and height tend to be correlated, this additional adjustment for height results in an odd phenotype that becomes hard to interpret.”

Authors Reply: We agree with the reviewer that Pulit et al. HMG 2019⁴ meta-analysis of WHRadjBMI contains a useful, systematic interrogation of collider bias. We made an effort to perform similar analyses to conclude that the risk of collider bias with BMI and height impacting these results is low.

The relevant passages from Pulit et al. HMG 2019⁴ followed by the analyses carried out in this study are shown below:

Pulit et al. HMG 2019:

“To investigate the potential for collider bias resulting from conditioning WHR on BMI, we investigated the behavior of WHRadjBMI-associated SNPs in GWAS of WHR (without adjustment for BMI) and BMI alone. We found that the majority of WHRadjBMI signals identified has genuine effect on body shape and that any bias caused by adjusting WHR for a correlated covariate (14,15) (that is, BMI) was minimal. Of the 346 index variants, 311 associated with stronger standard deviation effect sizes for WHR (unadjusted) than with standard deviation effect sizes for BMI (Supplementary Material, Table 3 and Supplementary Material, Fig. 4). This observation also indicates that the WHR association is unlikely to be secondary to the known effect of higher BMI resulting in higher WHR.”

This study:

We evaluated the fraction of lead SNPs ($P < 5 \times 10^{-8}$) for VATadj, ASATadj, and GFATadj that had stronger effect sizes for the unadjusted fat depot compared to effect sizes for BMI or height. We found that the majority of SNPs associated with adjusted fat depots were more strongly associated with the unadjusted fat depot than either of BMI or height (71-98%). This observation indicates that most genetic associations are unlikely to be secondary to collider bias with BMI or height.

Table The majority of lead SNPs identified with VATadj, ASATadj, and GFATadj are more strongly associated with the unadjusted fat depot than BMI or height			
	Lead SNPs	Lead SNPs where effect size for unadjusted fat depot is greater than BMI effect size	Lead SNPs where effect size for unadjusted fat depot is greater than height effect size

VAT adjusted for BMI and Height (VATadj)	30	26 (87%)	24 (80%)
ASAT adjusted for BMI and Height (ASATadj)	21	18 (86%)	15 (71%)
GFAT adjusted for BMI and Height (GFATadj)	54	53 (98%)	52 (96%)

Of note, Supplementary Figures SM1-3 in the Supplementary Methods plot each lead SNP and graphically illustrate those that are at higher risk of collider bias with BMI or height.

Pulit et al. HMG 2019:

“Furthermore, the common SNP associated with the largest known effect on BMI, that in the FTO gene (16), was not associated with WHRadjBMI (rs1421085, $P = 0.40$) despite a very strong association with WHR ($P = 4 \times 10^{-118}$).”

This study:

Despite strong associations between rs1421085 and VAT ($P = 5.7 \times 10^{-10}$), ASAT ($P = 1.7 \times 10^{-22}$), and GFAT (5.8×10^{-12}), either nonsignificant or nominal significance was observed with VATadj ($P = 0.02$), ASATadj ($P = 0.006$), and GFATadj ($P = 0.08$).

Pulit et al. HMG 2019:

“Finally, carrying each additional (weighted) WHRadjBMI-raising allele was associated with an increase in WHRadjBMI of 0.0199 SD ($P = 6 \times 10^{-62}$; adjusted $R^2 = 4\%$), an increase in WHR of 0.011 SD ($P = 3 \times 10^{-20}$; adjusted $R^2 = 0.12\%$) and a decrease in BMI of 0.004 SD ($P = 1.4 \times 10^{-3}$; adjusted $R^2 = 0.13\%$) in our independent dataset, consistent with the results obtained from an unweighted polygenic score (Methods).”

This study:

Finally, we aimed to determine the effect of the VATadj, ASATadj, and GFATadj polygenic scores derived in this study on the corresponding metric, the corresponding unadjusted fat depot volume, BMI, and height. We found in each case that the polygenic score was significantly associated with the adjusted fat depot and the corresponding unadjusted fat depot, but not BMI or height. Taking GFATadj as an example, a 1-standard deviation increase in the polygenic score associated with increased GFATadj (beta = 0.27, $P = 5.9e-122$) and increased GFAT (beta = 0.15, $P = 2.5e-38$), but a null effect with BMI (beta = 0.02, $P = 0.15$) and height (beta = 0.02, $P = 0.10$).

Table SM5 Association of VATadj, ASATadj, and GFATadj polygenic scores with VATadj, ASATadj, GFATadj, unadjusted metrics, BMI, and height

PRS	Trait	Beta (95% CI)	P-value	Adjusted R2
VATadj	VATadj	0.24 (0.22-0.26)	4.8e-101	0.0577
	VAT	0.13 (0.11-0.16)	4.8e-33	0.0179
	BMI	-0.02 (-0.04-0.01)	0.13	0.0001
	Height	-0.01 (-0.03-0.01)	0.54	0.0000
ASATadj	ASATadj	0.19 (0.17-0.21)	3.9e-62	0.0355
	ASAT	0.08 (0.06-0.11)	6.0e-14	0.0070
	BMI	0.00 (-0.02-0.02)	0.91	-0.0002
	Height	0.00 (-0.02-0.02)	0.78	-0.0001
GFATadj	GFATadj	0.27 (0.24-0.29)	5.9e-122	0.0703
	GFAT	0.15 (0.12-0.17)	2.5e-38	0.0210
	BMI	0.02 (-0.01-0.04)	0.15	0.0001
	Height	0.02 (0.00-0.04)	0.1	0.0003

Results reported here are from the 20% holdout set that was used to determine performance of polygenic scores. For all of VATadj, ASATadj, and GFATadj, the optimal set of LDpred2 hyperparameters in the validation set were $p = 0.0056$, $h^2 = 0.7$, $\text{sparse} = \text{FALSE}$ (Supplementary Table S22). To report performance metrics, each polygenic score was first adjusted for the first 10 PCs of genetic ancestry. Each PC-residualized polygenic score was then used to predict the trait of interest in a model that was adjusted for age at the time of imaging, sex, and the first 10 PCs of genetic ancestry. Betas correspond to sex-specific standard deviations per 1-standard deviation of the polygenic score. The adjusted R2 corresponds to R2 of the full model minus R2 of a model containing only covariates.

Regarding the interpretability of adjusting for BMI and height:

We have copied the below passage from the Supplementary Methods that discusses this issue:

Initially motivated by seminal work on waist-hip ratio adjusted for BMI led by the GIANT consortium, we started by examining the properties of VAT, ASAT, and GFAT adjusted for BMI (but not height).⁵ While genetic correlation with BMI was markedly reduced as desired, we noted that this adjustment introduced a significant genetic correlation with height (r_g ranging from 0.29 - 0.67) (**Table SM1 below**). As an example, GFAT adjusted for BMI (but not height) associated with rs67807996 ($P = 4.1 \times 10^{-14}$) and rs59985551 ($P = 2.1 \times 10^{-13}$) which have previously been identified as height-associated variants.^{6,7}

A similar phenomenon has previously been noted with waist circumference adjusted for BMI (WCadjBMI) and hip circumference (HIPadjBMI) adjusted for BMI in work led by the GIANT consortium:

“In contrast to WHRadjBMI, which has almost no genetic correlation with height ($r_g < 0.04$), WCadjBMI ($r_g = 0.42$) and HIPadjBMI ($r_g = 0.82$) have moderate genetic correlations with height. These data suggest that some, but not all, WCadjBMI and HIPadjBMI loci would be associated with height.”⁵

Accordingly, one of the height-associated variants noted above – rs59985551 – has also been associated with WCadjBMI and HIPadjBMI.⁸

By additionally adjusting for height, VAT adjusted for BMI and height (VATadj), ASATadj, and GFATadj achieved near height-independence (r_g ranging from -0.04 - 0.02) as desired. This strategy is consistent with the goal of this study to nominate genetic variants associated with “local adiposity” – i.e. genetic variants that influence adipose tissue volume in specific fat depots independent of the “overall size” of an individual. Of note, adjustment of each fat depot for BMI and height led to values that were nearly identical – both in terms of observational and genetic correlation – to adjusting each fat depot for weight and height. This latter strategy has previously been used to adjust CT-derived pericardial fat prior to genetic association.^{9,10}

Manuscript Change(s): We have modified the Supplementary Methods to include the following passage and **Table SM5**:

“Finally, we aimed to determine the effect of the VATadj, ASATadj, and GFATadj polygenic scores derived in this study on the corresponding metric, the corresponding unadjusted fat depot volume, BMI, and height. We found in each case that the polygenic score was significantly associated with the adjusted fat depot and the corresponding unadjusted fat depot, but not BMI or height. Taking GFATadj as an example, a 1-standard deviation increase in the polygenic score associated with increased GFATadj (beta = 0.27, $P = 5.9e-122$) and increased GFAT (beta = 0.15, $P = 2.5e-38$), but a null effect with BMI (beta = 0.02, $P = 0.15$) and height (beta = 0.02, $P = 0.10$).”

- Response to “The authors use PGSs to quantify/visualize the importance (or not) of the genetic contribution to individuals with extreme values. While a 3.8 fold greater likelihood seems impressive, ... That way, readers can interpret themselves how close (or far) we are from sing genetics in predicting complex traits.”

I must have missed the answer to me comment in your reponse. What is the misclassification. Or, what is your answer to this question “what percentage of those with a high GFATadjBMI values (or others) do not have a high PGS (e.g. in top 5%).” ?

You did not provide any explanation with the figures, but the way I read them is that of all the individuals with a high PGS, 10.5% has indeed a high VATadjBMI (and height), whereas 89.5% does not ?

Thus, there is an enormous misclassification when using the PGS to identify individuals at high risk. This needs to be made very explicit to provide a fair message to the readers. Currently, there is no mention of the misclassification.

Authors Reply: We agree with the reviewer that the predictive performance of the polygenic scores in this study should not be overstated. We make an effort to highlight that this enrichment does not imply that polygenic scores alone are a highly discriminatory test for each local adiposity trait by (1) including the variance explained by each polygenic score and (2) highlighting the absolute fraction of individuals in the top tail of GFATadj with a high GFATadj polygenic score (section copied below):

“To ensure no overlap between summary statistics and tested individuals, CVAS was conducted using a randomly selected 70% of participants. An additional 10% of participants was used as training data to select optimal LDPred2 hyperparameters and the remaining 20% of participants were held out for testing. In the test set, **VATadj, ASATadj, and GFATadj polygenic scores explained 5.8%, 3.6%, and 7.0% of the corresponding trait variance**, respectively (Supplementary Table S22-S23). Participants at the tails of the distribution for any of the three local adiposity traits were enriched in extreme polygenic scores – for example, participants in the top 5% of the GFATadj distribution were nearly four times as likely to have a GFATadj polygenic score in the top 5% of the distribution (**14.8% vs. 4.4%**; OR = 3.81; 95% CI: 2.76-5.17) (Figure 6; Supplementary Figure S14). Conversely, individuals with less than the 5th percentile of GFATadj were over three times as likely to have a GFATadj polygenic score less than the 5th percentile (**14.3% vs. 4.7%**; OR = 3.36; 95% CI: 2.32-4.77).”

To further address this reviewer’s concern, we have added a new **Supplementary Figure S14** that demonstrates the full distribution of polygenic scores that are possible in the phenotypic tails of VATadj, ASATadj, and GFATadj.

Manuscript Change(s): We have added a new **Supplementary Figure S14**.

Figure S14 Visualizing the relationship between VATadj, ASATadj, and GFATadj and their polygenic scores at the tails of the distributions

And made reference to this figure in the Results section:

“Participants at the tails of the distribution for any of the three local adiposity traits were enriched in extreme polygenic scores – for example, participants in the top 5% of the GFATadj distribution were nearly four times as likely to have a GFATadj polygenic score in the top 5% of the distribution (14.8% vs. 4.4%; OR = 3.81; 95% CI: 2.76-5.17) (Figure 6; **Supplementary Figure S14**). Conversely, individuals with less than the 5th percentile of GFATadj were over three times as likely to have a GFATadj polygenic score less than the 5th percentile (14.3% vs. 4.7%; OR = 3.36; 95% CI: 2.32-4.77). These findings suggest that polygenic inheritance plays an important role in fat distribution, and that polygenic scores could feasibly be used to enrich cohorts for individuals with extreme imaging phenotypes.”

1. Meyer HV, Dawes TJW, Serrani M, et al. Genetic and functional insights into the fractal structure of the heart. *Nature* 2020;584(7822):589–94.
2. Pirruccello JP, Bick A, Wang M, et al. Analysis of cardiac magnetic resonance imaging in 36,000 individuals yields genetic insights into dilated cardiomyopathy. *Nat Commun* 2020;11(1):2254.
3. Rask-Andersen M, Karlsson T, Ek WE, Johansson Å. Genome-wide association study of body fat distribution identifies adiposity loci and sex-specific genetic effects. *Nat Commun* 2019;10(1):339.
4. Pulit SL, Stoneman C, Morris AP, et al. Meta-analysis of genome-wide association studies for body fat distribution in 694 649 individuals of European ancestry. *Hum Mol Genet* 2019;28(1):166–74.
5. Shungin D, Winkler TW, Croteau-Chonka DC, et al. New genetic loci link adipose and insulin biology to body fat distribution. *Nature* 2015;518(7538):187–96.
6. Rieger S, McDaid A, Kutalik Z. Evaluation and application of summary statistic imputation to discover new height-associated loci. *PLoS Genet* 2018;14(5):e1007371.
7. Kichaev G, Bhatia G, Loh P-R, et al. Leveraging Polygenic Functional Enrichment to Improve GWAS Power. *Am J Hum Genet* 2019;104(1):65–75.
8. Christakoudi S, Evangelou E, Riboli E, Tsilidis KK. GWAS of allometric body-shape indices in UK Biobank identifies loci suggesting associations with morphogenesis, organogenesis, adrenal cell renewal and cancer. *Sci Rep* 2021;11(1):10688.
9. Chu AY, Deng X, Fisher VA, et al. Multiethnic genome-wide meta-analysis of ectopic fat depots identifies loci associated with adipocyte development and differentiation. *Nat Genet* 2017;49(1):125–30.
10. Fox CS, White CC, Lohman K, et al. Genome-wide association of pericardial fat identifies a unique locus for ectopic fat. *PLoS Genet* 2012;8(5):e1002705.

REVIEWER COMMENTS

Reviewer #2 (Remarks to the Author):

The authors have addressed all the comments made in the last round of review.

Minor suggestions:

- The analysis of collider bias is restricted to the methods/supplement. I would suggest at least adding a couple of sentences to the results section summarising to these analyses. I would also suggest flagging the lead SNPs in Table S4 that might be particularly biased to warn potential data users.

Response to Referees' Comments for NCOMMS-21-45422-T
"Inherited basis of visceral, abdominal subcutaneous, and gluteofemoral fat depots"

Reviewer #2 (Remarks to the Author):

The authors have addressed all the comments made in the last round of review.

Authors' Reply: Thank you for your helpful comments.

Minor suggestions:

- The analysis of collider bias is restricted to the methods/supplement. I would suggest at least adding a couple of sentences to the results section summarising to these analyses. I would also suggest flagging the lead SNPs in Table S4 that might be particularly biased to warn potential data users.

Authors' Reply: We agree that some discussion of the risk of collider bias is warranted in the Results.

Regarding the addition to **Table S4**, the current submission contains in **Table S26** the effect sizes of the unadjusted fat depot, BMI, and height, so that the reader may have complete context for each locus. To facilitate this, we have added a sentence to the caption of **Table S4**, directing the reader to **Table S26**.

Manuscript Change(s): In response, we have added a new section in the Results near the first mention of VATadj, ASATadj, and GFATadj to bring the potential issue of collider bias to the attention of the reader and direct them to the Supplementary Methods for a more complete interrogation:

"We tested VATadj, ASATadj, and GFATadj for possible collider bias with BMI or height and found minimal or no evidence of such bias for the majority of genome-wide significant loci (Supplementary Methods). For example, 87% of VATadj, 86% of ASATadj, and 98% of GFATadj genome-wide significant loci had stronger effect size for the unadjusted fat depot volume compared to BMI, comparable to the 90% of WHRadjBMI loci that met analogous criteria in a recent meta-analysis.¹²"

We have also added the following sentence for the caption of **Table S4**:

"For VATadj, ASATadj, and GFATadj results, **effect sizes for unadjusted fat depots, BMI, and height are included in Supplementary Table S26.**"